# Dual-modal metabolic analysis reveals hypothermia-reversible uncoupling of oxidative phosphorylation in neonatal brain hypoxia-ischemia

**Naidi Sun**[1,2†‡]**, Yu-Yo Sun**[3,4,5†]**, Rui Cao**[2]**, Hong-Ru Chen**[4,5,6]**, Yiming Wang**[2]**, Elizabeth Fugate**[7]**, Marchelle R Smucker**[4,5]**, Yi-Min Kuo**[8,9]**, Ellen P Grant**[10]**, Diana M Lindquist**[7]**, Chia-Yi Kuan**[4,5*]**, Song Hu**[1*]

[1]Department of Biomedical Engineering, Washington University in St. Louis, St. Louis, United States; [2]Department of Biomedical Engineering, University of Virginia, Charlottesville, United States; [3]Institute of BioPharmaceutical Sciences, National Sun Yat-Sen University, Kaohsiung, Taiwan; [4]Department of Neuroscience, University of Virginia, Charlottesville, United States; [5]Center for Brain Immunology and Glia (BIG), University of Virginia, Charlottesville, United States; [6]Department of Life Sciences and Institute of Genome Sciences, National Yang Ming Chiao Tung University, Taipei, Taiwan; [7]Department of Radiology, Cincinnati Children's Hospital Medical Center, Cincinnati, United States; [8]Department of Anesthesiology, Taipei Veterans General Hospital, Taipei, Taiwan; [9]Department of Anesthesiology, College of Medicine, National Yang Ming Chiao Tung University, Taipei, Taiwan; [10]Fetal-Neonatal Neuroimaging and Developmental Science Center, Boston Children's Hospital, Boston, United States

**\*For correspondence:**
ck6cb@virginia.edu (C-YK);
songhu@wustl.edu (SH)

[†]These authors contributed equally to this work

**Present address:** [‡]Key Laboratory of Brain Health Intelligent Assessment and Intervention of the Ministry of Education, School of Medical Technology, Beijing Institute of Technology, Beijing, China

**Competing interest:** The authors declare that no competing interests exist.

## eLife Assessment

This is an **important** study that utilized in vivo optical measurements of the cortical metabolic rate of O2 and blood flow, as well as measurements in isolated mitochondria to assess the uncoupling of the oxidative phosphorylation due to hypoxia-ischemia injury of the neonatal brain, and effects of the hypothermia treatment. The combination of state-of-the-art optical measurements, mitochondrial assays, and the use of various control experiments provides **convincing** evidence for the derived conclusions. This work will be of interest to those in the mitochrondrial metabolomics, brain injury and hypoxia fields.

**Abstract** Hypoxia-ischemia (HI), which disrupts the oxygen supply-demand balance in the brain by impairing blood oxygen supply and the cerebral metabolic rate of oxygen ($CMRO_2$), is a leading cause of neonatal brain injury. However, it is unclear how post-HI hypothermia helps to restore the balance, as cooling reduces $CMRO_2$. Also, how transient HI leads to secondary energy failure (SEF) in neonatal brains remains elusive. Using photoacoustic microscopy, we examined the effects of HI on $CMRO_2$ in awake 10-day-old mice, supplemented by bioenergetic analysis of purified cortical mitochondria. Our results show that while HI suppresses ipsilateral $CMRO_2$, it sparks a prolonged $CMRO_2$-surge post-HI, associated with increased mitochondrial oxygen consumption, superoxide emission, and reduced mitochondrial membrane potential necessary for ATP synthesis—indicating oxidative phosphorylation (OXPHOS) uncoupling. Post-HI hypothermia prevents the $CMRO_2$-surge

by constraining oxygen extraction fraction, reduces mitochondrial oxidative stress, and maintains ATP and N-acetylaspartate levels, resulting in attenuated infarction at 24 hr post-HI. Our findings suggest that OXPHOS-uncoupling induced by the post-HI $CMRO_2$-surge underlies SEF and blocking the surge is a key mechanism of hypothermia protection. Also, our study highlights the potential of optical $CMRO_2$ measurements for detecting neonatal HI brain injury and guiding the titration of therapeutic hypothermia at the bedside.

## Introduction

The human brain constitutes only 2% of the body mass, but utilizes ~20% of body's oxygen consumption (*Rolfe and Brown, 1997*). Disruptions in cerebral blood flow and oxygen supply cause a spectrum of brain injuries, including adult ischemic stroke and neonatal hypoxic-ischemic encephalopathy (HIE; *Allen and Brandon, 2011*). In adult ischemic stroke, a >60% reduction in the cerebral metabolic rate of oxygen ($CMRO_2$) is indicative of looming infarction (*Lee et al., 2003*; *Lin and Powers, 2018*). However, the impacts of HIE on cerebral oxygen metabolism are far less certain due to the challenge of measuring $CMRO_2$ in infant brains by magnetic resonance imaging (MRI) or positron emission tomography (PET). Studies in animal models of HIE have revealed an initial recovery of brain adenosine triphosphate (ATP) levels during a short latent period after hypoxia-ischemia (HI), followed by a precipitous decline known as the secondary energy failure (SEF) that signifies looming infarction (*Blumberg et al., 1997*; *Yager et al., 1992*). However, the mechanisms of post-HI SEF and how therapeutic hypothermia protects neonatal brains against HI injury remain uncertain (*Gunn et al., 2017*). Also, it is unclear whether $CMRO_2$ measurements with recently developed bedside optical instruments, free of MRI's susceptibility to infant motion and PET's radiotoxicity, can detect HIE brain injury in a manner similar to adult ischemic stroke (*De Carli et al., 2019*; *Dehaes et al., 2014*; *Ferradal et al., 2017*; *Jain et al., 2014*).

Mitochondrial bioenergetics is vital to cellular functions and survival, and dysregulated oxidative-phosphorylation (OXPHOS) may promote neonatal HI brain injury (*Niatsetskaya et al., 2012*). While the phosphorylation efficiency of mitochondrial respiration (P/O ratio) remains stable over a wide range of substrate concentrations, it declines under hypoxia through a process known as OXPHOS-uncoupling (*Kramer and Pearlstein, 1983*), which is indicated by the rise of oxygen consumption despite a reduction of the mitochondrial membrane potential that is needed for ATP synthesis (*Kristián and Siesjö, 1998*). Although OXPHOS-uncoupling has been implicated in decoupling the contractive strength from oxidative metabolism after cardiac ischemia (*Benzi and Lerch, 1992*; *Juhaszova et al., 2004*), its role in HIE is unknown. We hypothesized that OXPHOS-uncoupling may be associated with mitochondrial injury in HIE, serving as the cause of post-HI SEF and a target for therapeutic hypothermia.

To test this hypothesis, we applied a head-restrained photoacoustic microscopy (PAM) technique to measure $CMRO_2$ during and immediately after HI in awake 10-day-old (P10) mice. Using multi-parametric PAM, we simultaneously acquired high-resolution images of cerebral blood flow (CBF), the oxygen saturation of hemoglobin ($sO_2$), and the total concentration of hemoglobin ($C_{Hb}$), which can be combined to calculate $CMRO_2$ in absolute values (*Figure 1A* and see Methods for details of the PAM system). The use of awake mouse neonates avoided the confounding effects of anesthesia on CBF and $CMRO_2$ (*Cao et al., 2017*; *Gao et al., 2017*; *Sciortino et al., 2021*; *Slupe and Kirsch, 2018*). In addition, we measured the oxygen consumption rate (OCR), reactive oxygen species (ROS), and the membrane potential of mitochondria that were immediately purified from the same cortical area imaged by PAM. This dual-modal analysis enabled a direct comparison of cerebral oxygen metabolism and cortical mitochondrial respiration in the same animal. Moreover, we compared the effects of therapeutic hypothermia on oxygen metabolism and mitochondrial respiration and correlated the extent of $CMRO_2$ reduction with the severity of infarction at 24 hr after HI. Our results suggest that blocking HI-induced OXPHOS-uncoupling is an acute effect of hypothermia and that optical detection of $CMRO_2$ may have clinical applications in HIE.

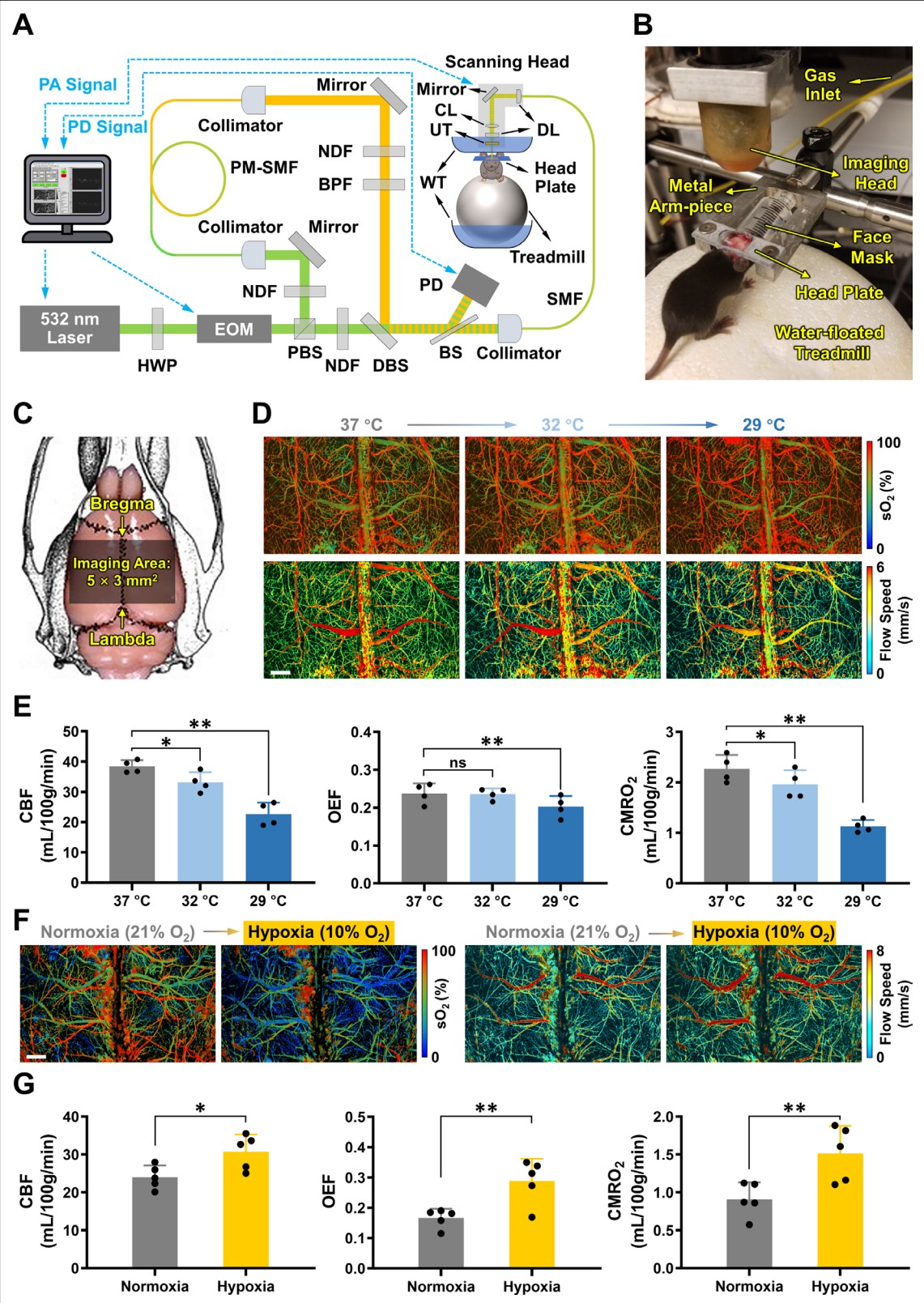

**Figure 1.** Multi-parametric photoacoustic microscopy (PAM) of hemodynamic and oxygen-metabolic responses of uninjured neonatal mouse brains to hypothermia or hypoxia. (**A**) Schematic of the head-restrained multi-parametric PAM system. PA, photoacoustic; PD, photodiode; HWP, half-wave plate; EOM, electro-optical modulator; PBS, polarizing beam splitter; NDF, neutral-density filter; PM-SMF, polarization-maintaining single-mode fiber; BPF, band-pass filter; DBS, dichroic beam splitter; BS, beam sampler; SMF, single-mode fiber; DL, doublet; CL, correction lens; UT, ring-shaped ultrasonic

*Figure 1 continued on next page*

*Figure 1 continued*

transducer; WT, water tank. (**B**) Photograph of the placement of an awake 10-day-old (P10) mouse wearing the head plate in the PAM system. Note that the water tank is removed to better show the mouse and related parts for head-restrained awake-brain imaging. (**C**) Illustration of the imaging field (5×3 mm$^2$) that covers both hemicortices between the Bregma and Lambda on the skull of mouse neonates. (**D**) Multi-parametric PAM images of the oxygen saturation of hemoglobin (sO$_2$) and blood flow speed in an awake P10 mouse, with the skull temperature set at 37, 32, and 29 °C, respectively. Scale bar: 500 µm. (**E**) Hemodynamic and oxygen-metabolic responses of the neonatal mouse cortex to different skull temperatures, including cerebral blood flow (CBF), oxygen extraction fraction (OEF), and the cerebral metabolic rate of oxygen (CMRO$_2$). Gray bars: 37 °C; light blue bars: 32 °C; dark blue bars: 29 °C. One-way ANOVA was performed, and data are presented as mean ± standard deviation (n = 4). ns, no significance; *, p < 0.05; **, p < 0.01. (**F**) Multi-parametric PAM images of sO$_2$ and blood flow speed in an anesthetized P10 mouse under normoxia (inhaled oxygen concentration: 21%) versus hypoxia (inhaled oxygen concentration: 10%). Scale bar: 500 µm. (**G**) Hemodynamic and oxygen-metabolic responses of the neonatal mouse cortex to normoxia vs. hypoxia, including CBF, OEF, and CMRO$_2$. Gray bars: normoxia; orange bars: hypoxia. A Student t-test was performed, and data are presented as mean ± standard deviation (n = 5). *, p < 0.05; **, p < 0.01.

The online version of this article includes the following source data and figure supplement(s) for figure 1:

**Source data 1.** Source data for *Figure 1E*.

**Source data 2.** Source data for *Figure 1G*.

**Figure supplement 1.** Relationship between the water tank temperature and the mouse skull temperature.

## Results

### Effects of cooling and hypoxia on CMRO$_2$ in uninjured mouse neonates

First, we studied the effects of hypothermia alone on neonatal CMRO$_2$. As shown in *Figure 1A and B*, an awake P10 mouse was placed between the PAM apparatus and a water-floated voluntary treadmill, with the temperature of the water tank above the mouse head adjusted to achieve normothermia or hypothermia based on the skull temperature (*Figure 1—figure supplement 1*). Note that while the mouse skull was exposed, no craniotomy was necessary. Multi-parametric PAM images over a 5×3 mm$^2$ region from the Bregma to Lambda in both cortices were acquired within 40 min (*Figure 1C*). Using this system, we assessed the effects of hypothermia (from 37 °C to 32 °C to 29 °C of the mouse skull temperature) on CMRO$_2$ in uninjured mice (*Figure 1D*). Quantitative analysis revealed a decline in CBF from 38.37 mL/100 g/min at 37 °C to 33.18 mL/100 g/min at 32 °C (86% of the baseline) and to 22.65 mL/100 g/min at 29 °C (59% of the baseline; *Figure 1E*, left). In contrast, OEF remained unchanged at 32 °C and slightly declined to 85% of the baseline at 29 °C (*Figure 1E*, middle). Consequently, CMRO$_2$ declined from 2.27 mL/100 g/min at 37 °C to 1.97 mL/100 g/min at 32 °C (86% of the baseline) and further to 1.13 mL/100 g/min at 29 °C (50% of the baseline; *Figure 1E*, right). These results show that hypothermia reduces CMRO$_2$ in uninjured mouse neonates, primarily through the suppression of CBF.

Next, we studied the effects of hypoxia alone on neonatal CMRO$_2$. Exposing anesthetized P10 mice to hypoxia (10% O$_2$) resulted in an increase of CMRO$_2$ (from 0.91 to 1.51 mL/100 g/min), which was attributed to the rise in both CBF (from 24.04 to 30.74 mL/100 g/min) and OEF (from 0.17 to 0.29; *Figure 1F and G*). While this observation of greater oxygen consumption under a lower oxygen tension is seemingly counterintuitive, it aligns with the previous reports of higher CMRO$_2$ and cytochrome c oxidase activity in mild hypoxia (*Tsuji et al., 1995*; *Vestergaard et al., 2016*). This phenomenon is presumably due to an increased cerebral energy demand under hypoxia.

### Effects of combined HI on CMRO$_2$ and mitochondrial respiration

Then, we studied the effects of combined HI under the Vannucci model, which involves unilateral common carotid artery (CCA) occlusion and transient hypoxia (*Kuan et al., 2021*; *Yang et al., 2009*). Using time-lapse PAM, we compared the dynamic changes of CBF, OEF, and CMRO$_2$ in the two hemicortices from unilateral CCA-ligation to combined HI (40 min), and for 2 hr upon returning to normoxia in awake P10 mice (*Figure 2A–D* and *Supplementary file 1*).

Before the onset of hypoxia, CCA-ligation alone did not provoke a significant difference in CMRO$_2$ between the two hemicortices (1.56 *vs* 1.26 mL/100 g/min; p-value, 0.6107; the first pair of red and green points in *Figure 2D*). When hypoxia was initiated, however, the contralateral (CL) cortex had a significant increase in CMRO$_2$, similar to the responses in uninjured mice, while the CCA-ligated counterpart showed a marked reduction in CMRO$_2$ (0.78 *vs* 2.37 mL/100 g/min; p-value, 0.0011; the second pair of red and green points in *Figure 2D*). Upon switching back to normoxia, CMRO$_2$ in the

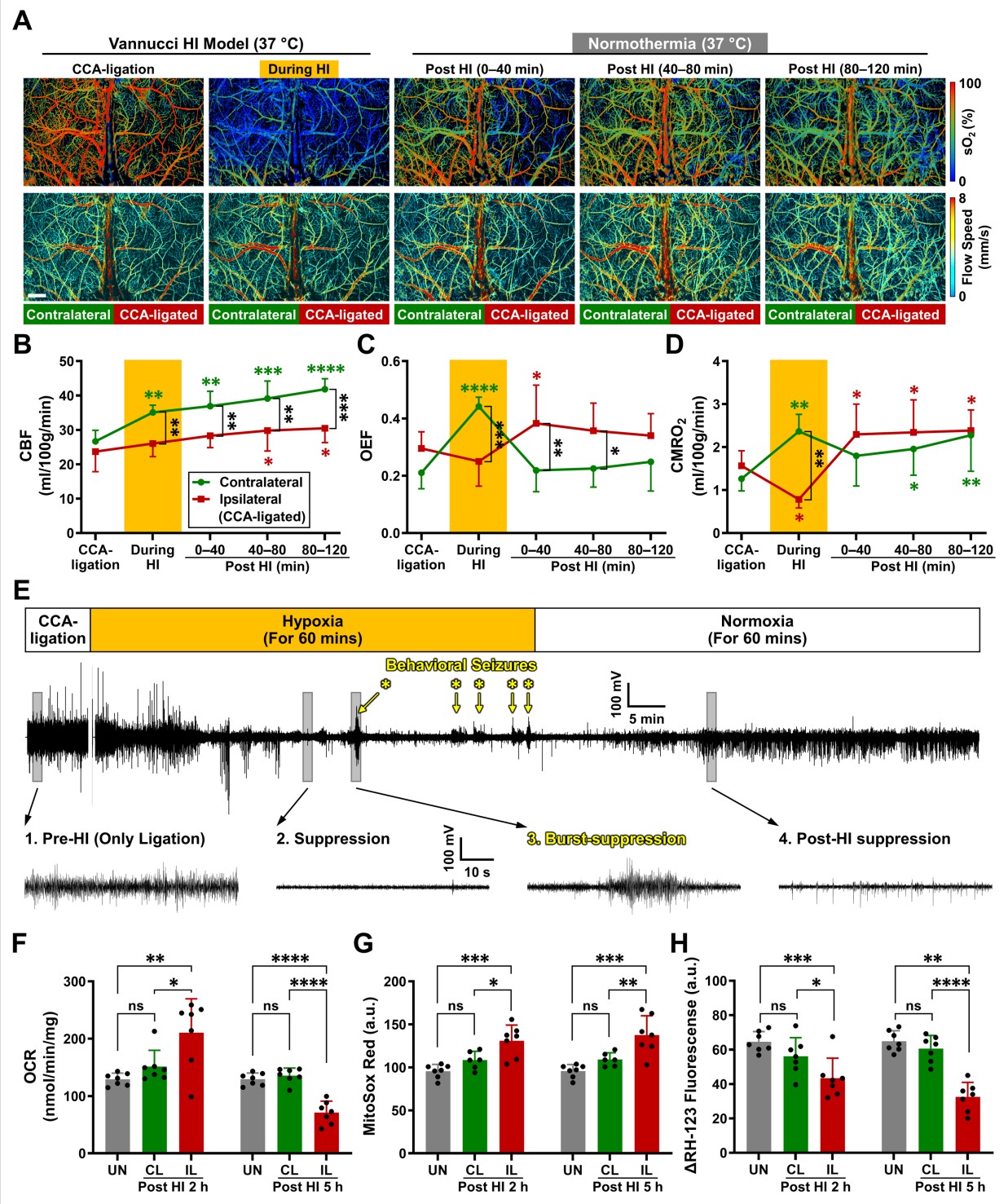

**Figure 2.** Effects of combined hypoxia-ischemia (HI) on cerebral hemodynamics, oxygen metabolism, and mitochondrial bioenergetics in mouse neonates under normothermia. (**A**) Time-lapse PAM images of the oxygen saturation of hemoglobin ($sO_2$) and blood flow speed in an awake P10 mouse during unilateral CCA-ligation, combined HI, as well as 0–40, 40–80, and 80–120 minutes post-HI. Scale bar: 500 µm. (**B–D**) Hemodynamic and oxygen-metabolic responses of the contralateral (green line and symbols) and ipsilateral (red line and symbols) cortices to the Vannucci HI Model under normothermia, including (**B**) cerebral blood flow (CBF), (**C**) oxygen extraction fraction (OEF), and (**D**) the cerebral metabolic rate of oxygen ($CMRO_2$). Green and red asterisks indicate the statistical significance, if any, over the baseline (i.e. CCA-ligation alone) values measured in the same animal, while black asterisks indicate the statistical significance, if any, between the measurements in the two hemicortices at the same time point. Two-way ANOVA was performed, and data are presented as mean $\pm$ standard deviation (n = 5). *, $p < 0.05$; **, $p < 0.01$; ***, $p < 0.001$; ****, $p < 0.0001$.

*Figure 2 continued on next page*

*Figure 2 continued*

(**E**) Electroencephalography (EEG) recording of the ipsilateral cortex in a P10 mouse right after the unilateral CCA-ligation, 60-min hypoxia (inhaled oxygen concentration: 10%), and 60-min normoxia (inhaled oxygen concentration: 21%). Zoom-in views of the boxed regions show characteristic EEG patterns, including 1. pre-HI baseline, 2. Suppression, 3. burst-suppression correlated with seizure behaviors (highlighted by yellow asterisks), and 4. post-HI suppression. n=6. (**F–H**) Comparison of the mitochondrial parameters acquired in the uninjured (UN, gray bars), contralateral (CL, green bars), and ipsilateral (IL, red bars) cortices at 2 or 5 hr post-HI, including (**F**) oxygen consumption rate (OCR), (**G**) MitoSox Red, and (**H**) ΔRH-123 fluorescence. One-way ANOVA was performed, and data are presented as mean ± standard deviation (n = 7). ns, no significance; *, $p < 0.05$; **, $p < 0.01$; ***, $p < 0.001$; ****, $p < 0.0001$.

The online version of this article includes the following source data and figure supplement(s) for figure 2:

**Source data 1.** Source data for *Figure 2B*.

**Source data 2.** Source data for *Figure 2C*.

**Source data 3.** Source data for *Figure 2D*.

**Source data 4.** Source data for *Figure 2F*.

**Source data 5.** Source data for *Figure 2G*.

**Source data 6.** Source data for *Figure 2H*.

**Figure supplement 1.** Oxygen consumption of mitochondria isolated from the uninjured (UN), contralateral (CL), and ipsilateral (IL) cortex at (**A**) 2 hr or (**B**) 5 hr post-HI.

**Figure supplement 1—source data 1.** Source data for *Figure 2—figure supplement 1A*.

**Figure supplement 1—source data 2.** Source data for *Figure 2—figure supplement 1B*.

CL cortex quickly returned to the pre-hypoxia level and rose slowly afterwards, while $CMRO_2$ in the ipsilateral (IL) cortex rapidly rebounded above the pre-hypoxia level and remained elevated for at least 2 hr (the last three red and green points in *Figure 2D*).

To determine the causes of the post-HI $CMRO_2$-surge, we recorded electroencephalography (EEG) of the P10 mouse before, during, and after HI to test whether the surge was due to status epilepticus (*Lin and Powers, 2018*). Although showing seizure-like behaviors and a few spike-wave discharges on top of the EEG suppression during HI (*Figure 2E*, hypoxia), mice only showed a gradual recovery from the suppression without burst discharges on EEG post-HI (*Figure 2E*, normoxia). Thus, the post-HI $CMRO_2$-surge was not attributed to status epilepticus.

We then isolated the mitochondria from both hemicortices of injured mouse brains at 2 or 5 hr post-HI and from the uninjured (UN) mouse brains for in vitro analysis (see Methods for details of the mitochondrial function analysis). At 2 hr post-HI, mitochondria from the IL cortex showed increased OCR (*Figure 2F* and *Figure 2—figure supplement 1*) and superoxide (as measured by the MitoSox Red fluorescence, *Figure 2G*), but reduced mitochondrial membrane potential ($\psi_m$, as measured by the ΔRH-123 fluorescence, *Figure 2H*), compared to mitochondria in the CL cortex or the UN brain. By 5 hr post-HI, mitochondria in the IL cortex manifested a lower OCR and $\psi_m$, but an even higher superoxide emission, suggesting more substantial mitochondria injury (*Figure 2F–H*). In contrast, mitochondria in the CL cortex at both 2 and 5 hr post-HI showed comparable OCR, superoxide, and $\psi_m$ to those from the UN brain (*Figure 2F–H*). These results suggest that post-HI $CMRO_2$-surge can be attributed to OXPHOS-uncoupling and ROS-emission, which may promote mitochondrial injury.

## Effects of hypothermia treatment on HI-induced $CMRO_2$ surge and mitochondria injury

Next, we tested the effects of therapeutic hypothermia on $CMRO_2$ and mitochondrial functions. To mimic clinical settings, CCA-ligation and combined HI were performed at the room temperature, while hypothermia was initiated after HI by reducing the mouse skull temperature from 37 °C to 32 °C (*Figure 3A–D*).

Similar to *Figure 2D*, HI provoked a rise of $CMRO_2$ in the CL cortex and a decline of $CMRO_2$ in the IL cortex (*Figure 3D*). Notably, post-HI hypothermia blunted the $CMRO_2$-surge in the IL cortex, locking it at the pre-HI level throughout the 2 hr monitoring period (*Figure 3D* and *Supplementary file 2*). Comparison of the PAM measurements obtained under normothermia vs. hypothermia revealed that the chief effect of hypothermia was to block the rise of post-HI OEF rather than reducing

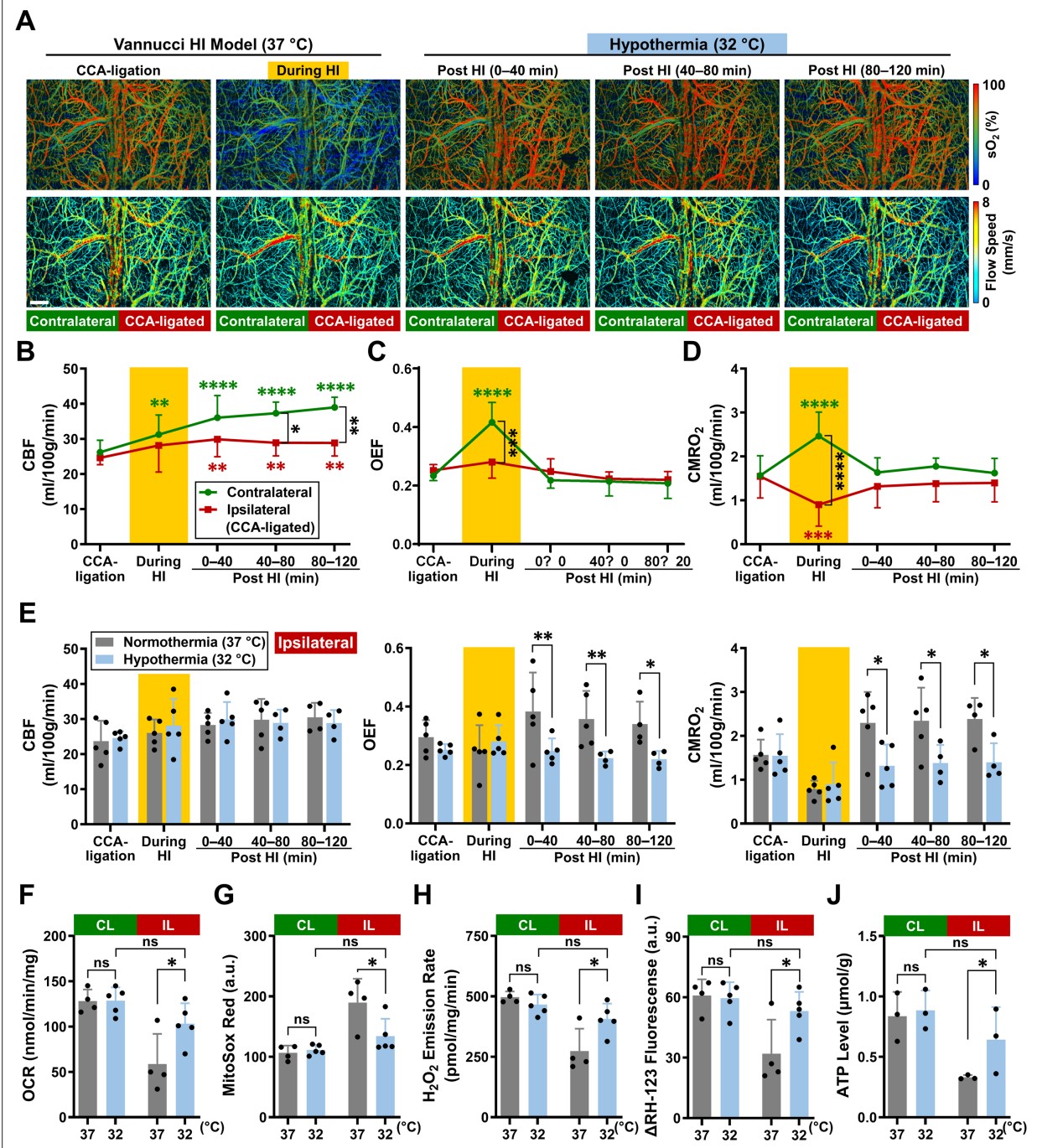

**Figure 3.** Effects of hypoxia-ischemia (HI) on cerebral hemodynamics, oxygen metabolism, and mitochondrial bioenergetics in mouse neonates under hypothermia *vs.* normothermia. (**A**) Time-lapse PAM images of the oxygen saturation of hemoglobin (sO$_2$) and blood flow speed in an awake P10 mouse during unilateral CCA-ligation, combined HI, as well as 0–40, 40–80, and 80–120 min post-HI. Scale bar: 500 µm. (**B–D**) Hemodynamic and oxygen-metabolic responses of the contralateral (green line and symbols) and ipsilateral (red line and symbols) cortices to the Vannucci HI Model under hypothermia, including (**B**) cerebral blood flow (CBF), (**C**) oxygen extraction fraction (OEF), and (**D**) the cerebral metabolic rate of oxygen (CMRO$_2$). Green and red asterisks indicate the statistical significance, if any, over the baseline (i.e. CCA-ligation alone) values measured in the same animal, while black asterisks indicate the statistical significance, if any, between the measurements in the two hemicortices at the same time point. (**E**) Comparison of CBF, OEF, and CMRO$_2$ responses of the ipsilateral cortex to the HI insult under normothermia (37 °C, gray bars) *vs* hypothermia (32 °C, light blue bars). For (**B–E**), two-way ANOVA was performed, and data are presented as mean ± standard deviation (n = 5). *, $p < 0.05$; **, $p < 0.01$; ***, $p < 0.001$; ****, $p < 0.0001$. (**F–I**) Comparison of the mitochondrial parameters acquired in the contralateral (CL, green bars) and ipsilateral (IL, red bars) cortices at 5 hr post-HI under normothermia (37 °C, gray bars) *vs* hypothermia (32 °C, light blue bars), including (**F**) oxygen consumption rate (OCR), (**G**) MitoSox Red, (**H**) H$_2$O$_2$ emission rate, and (**I**) ΔRH-123 fluorescence. Two-way ANOVA was performed, and data are presented as mean ± standard deviation

*Figure 3 continued on next page*

*Figure 3 continued*

(n = 4 or 5 for normothermia or hypothermia treatment group, respectively). ns, no significance; *, p < 0.05. (**J**) Comparison of the ATP concentrations measured in the CL and IL cortices at 6 hr post-HI under normothermia (37 °C, gray bars) *vs* hypothermia (32 °C, light blue bars). Two-way ANOVA was performed, and data are presented as mean ± standard deviation (n = 3). ns, no significance; *, p < 0.05. For (**F–J**), the in-vitro analyses were performed in mitochondria isolated from HI-injured mice with or without a 4 hr hypothermia treatment, followed by another (**F–I**) one hour or (**J**) two hours recovery in normothermia.

The online version of this article includes the following source data for figure 3:

**Source data 1.** Source data for *Figure 3B*.

**Source data 2.** Source data for *Figure 3C*.

**Source data 3.** Source data for *Figure 3D*.

**Source data 4.** Source data for *Figure 3E*.

**Source data 5.** Source data for *Figure 3F*.

**Source data 6.** Source data for *Figure 3G*.

**Source data 7.** Source data for *Figure 3H*.

**Source data 8.** Source data for *Figure 3I*.

**Source data 9.** Source data for *Figure 3J*.

CBF (*Figure 3E*), as observed in the uninjured neonates (*Figure 1E*). Similar to our results, a recent study also showed minimal reduction of post-HI CBF by hypothermia at 32 °C (*Buckley et al., 2015*).

We then tested whether therapeutic hypothermia protects mitochondria and prevents the onset of SEF, as previously postulated (*Gunn et al., 2017*). We isolated mitochondria from the CL and IL cortex, with or without the 4 hr hypothermia treatment for in-vitro analysis at 5 hr post-HI (i.e. 1 hr after the conclusion of the hypothermia treatment). As shown in *Figure 3F–I*, mitochondria isolated from the IL cortex without hypothermia treatment showed reduced OCR, higher superoxide, as well as lower hydrogen peroxide ($H_2O_2$) and mitochondria $\phi_m$, compared to those receiving hypothermia treatment or from the CL cortex. In contrast, mild hypothermia (32 °C) had no suppressive effects on OCR, superoxide and $H_2O_2$ emission, and $\phi_m$ of mitochondria in the CL cortex. We also compared the mitochondrial ATP levels at 6 hr post-HI and found a higher ATP level (0.64 µmol/g) in the IL cortex receiving hypothermia treatment than those without hypothermia (0.33 µmol/g, *Figure 3J*). These results confirmed that therapeutic hypothermia maintained mitochondrial OCR and $\phi_m$, reduced superoxide and $H_2O_2$ emission, and prevented post-HI SEF. In addition, in vivo proton MRS in infants with HIE has also shown a reduction in NAA, particularly in cases of severe injury (*Lally et al., 2019*). This reduction in NAA, observed in neonatal intensive care settings, reflects neuronal and axonal loss or dysfunction and serves as a biomarker for injury severity. The alignment between our ex vivo observations and in vivo MRS findings in clinical studies reinforces the translational relevance of our model for investigating metabolic disturbances in neonatal HIE.

## Using optically measured $CMRO_2$ to detect neonatal HI brain injury

Finally, we compared $CMRO_2$ and brain infarction with or without hypothermia treatment at 24 hr after HI. PAM measurements (*Figure 4A*) and tissue-level $CMRO_2$ mapping (*Figure 4B*) showed better preservation of oxygen metabolism in the IL cortex of hypothermia-treated mice, compared with those without the treatment. Notably, the marked reduction of $CMRO_2$ at 24 hr post-HI in non-treated mice (*Figure 4B*) was coupled to impaired triphenyl-tetrazolium chloride (TTC) staining, which corresponded to tissue infarction (*Figure 4C*). The TTC analysis confirmed the reduction of infarct size in mice receiving therapeutic hypothermia than those without (8.74 mm³ *vs.* 45.25 mm³, *Figure 4D*). Direct comparison of the hemodynamic and oxygen-metabolic parameters showed that post-HI hypothermia maintained CBF (41.13 mL/100 g/min), OEF (0.23), and $CMRO_2$ (2.12 mL/100 g/min) in the IL cortex to a level similar to those in the CL cortex (blue bars in *Figure 4E–G*). In contrast, mice without hypothermia treatment showed a marked reduction of OEF (0.04) and $CMRO_2$ (0.35 mL/100 g/min) in the IL cortex at 24 hours post-HI (gray bars in *Figure 4E–G*).

Additionally, the proton nuclear magnetic resonance (NMR) analysis showed significant reductions of total creatine (Cr/PCr), choline (Cho), glutamate and glutamine (Glx), and NAA in the IL cortex, compared to those in the CL cortex, in mice without hypothermia treatment (*Figure 4H* and gray

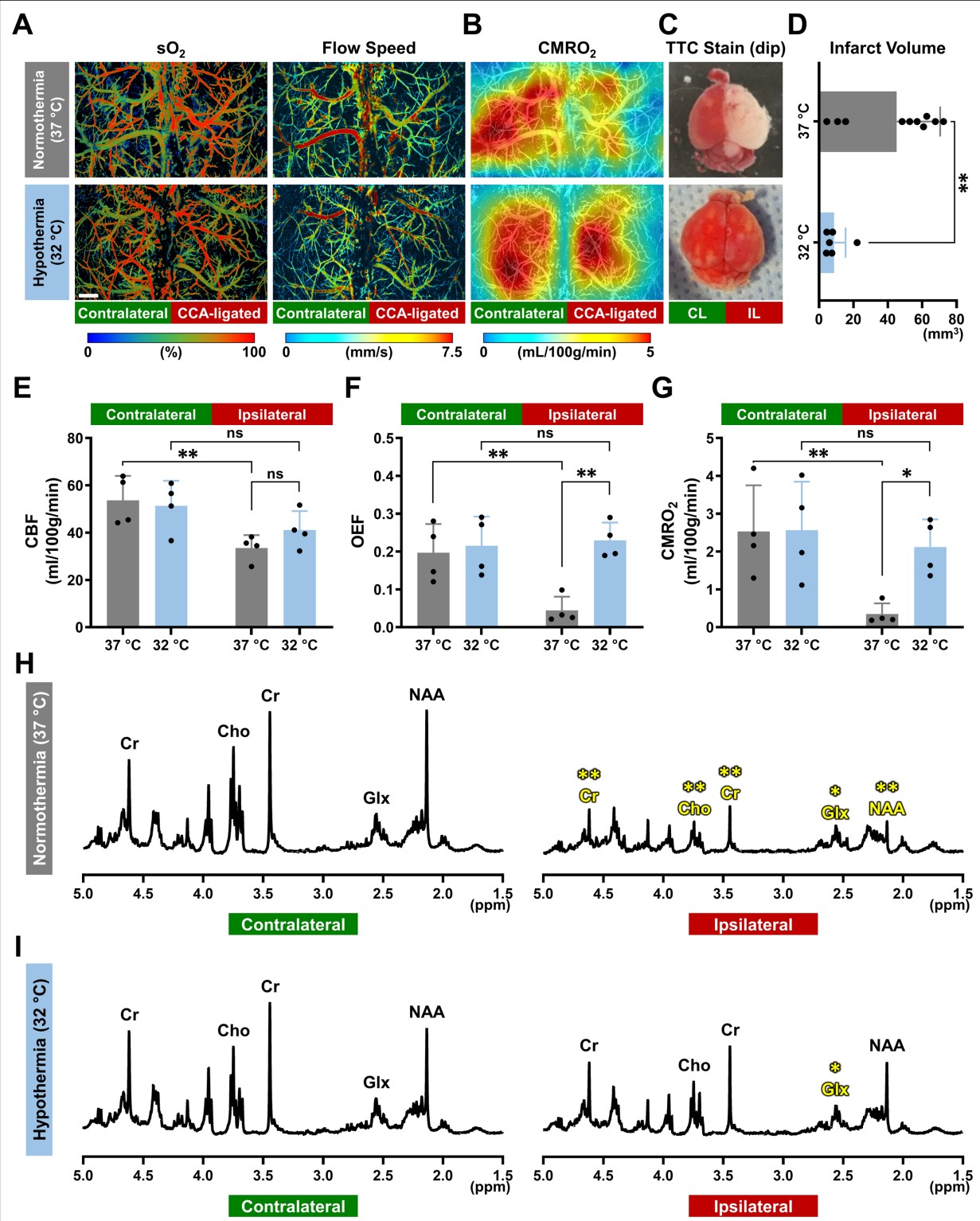

**Figure 4.** Correlation of CMRO$_2$ and post-HI brain infarction in mouse neonates at 24 hr. (**A–C**) Multi-parametric PAM images of (**A**) the oxygen saturation of hemoglobin (sO$_2$) and blood flow speed and (**B**) the cerebral metabolic rate of oxygen (CMRO$_2$), as well as (**C**) the triphenyl-tetrazolium chloride (TTC) analysis of the dissected brain after completion of the in vivo PAM imaging at 24 hr post-HI. The top and bottom rows are data acquired in the neonate brain treated with normothermia (37 °C, upper row) or hypothermia (32 °C, lower row) for 4 hr, respectively. CL, contralateral; IL, ipsilateral. Scale bar: 500 μm. (**D**) Comparison of the infarct volume quantified based on the TTC analysis (n=6 for the normothermia group and n=10 for the hypothermia group). A Student *t*-test was performed, and data are presented as mean ± standard deviation. **, p < 0.01. (**E–G**)

*Figure 4 continued on next page*

*Figure 4 continued*

Comparison of (**E**) cerebral blood flow (CBF), (**F**) oxygen extraction fraction (OEF), and (**G**) $CMRO_2$ acquired at 24 hr post-HI in the neonate brain treated with normothermia (37 °C, gray bars) or hypothermia (32 °C, light blue bars) for 4 hr. Two-way ANOVA was performed, and data are presented as mean ± standard deviation (n = 4). ns, no significance; *, $p < 0.05$; **, $p < 0.01$. (**H–I**) Characteristic proton-HRMAS spectra of the snap-frozen tissue from the contralateral and ipsilateral cortex extracted at 24 hr post-HI from HI-injured mice treated with (**H**) normothermia or (**I**) hypothermia. The spectra are scaled to the same peak height for myo-inositol. Cr, creatine and phosphocreatine; Cho, choline; Glx, glutamate and glutamine; NAA, N-acetyl aspartate. Two-way ANOVA was performed (n = 5 for normothermia and n=3 for hypothermia). *, $p < 0.05$; **, $p < 0.01$.

The online version of this article includes the following source data and figure supplement(s) for figure 4:

**Source data 1.** Source data for *Figure 4D*.

**Source data 2.** Source data for *Figure 4E*.

**Source data 3.** Source data for *Figure 4F*.

**Source data 4.** Source data for *Figure 4G*.

**Source data 5.** Source data for *Figure 4H*.

**Source data 6.** Source data for *Figure 4I*.

**Figure supplement 1.** [1]H-HRMAS MRS analysis of the effects of normothermia *vs.* hypothermia on brain metabolites at 24 hr post-HI.

**Figure supplement 1—source data 1.** Source data for *Figure 4—figure supplement 1*.

bars in *Figure 4—figure supplement 1*). In contrast, mice that received therapeutic hypothermia maintained similar levels of Cr/PCr, Cho, and NAA between the two hemicortices (*Figure 4I* and light blue bars in *Figure 4—figure supplement 1*). These results suggest that marked reduction of post-HI $CMRO_2$ signifies SEF and tissue infarction, similar to that in adult ischemic stroke (*Lee et al., 2003*; *Lin and Powers, 2018*).

## Discussion

Acute post-asphyxia HIE around the time of birth remains a major cause of neonatal death and lifelong neurological deficits. Clinical observations and animal studies suggest that brain cells initially recover from the insult during a short latent phase, typically lasting ~6 hr, but then enter a deterioration phase characterized by progressive failure of oxygen metabolism (i.e. SEF), seizures, and death (*Blumberg et al., 1997*; *Gunn et al., 2017*; *Yager et al., 1992*). Acute treatment with mild hypothermia (32–34 °C) prevents the onset of SEF in moderate HIE, but the mechanisms remain unclear for at least two reasons (*Gunn et al., 2017*). First, hypothermia normally inhibits instead of enhancing cerebral oxygen metabolism. Second, since mild hypothermia has little effect on the EEG activity (*Erecinska et al., 2003*), which does not support the common notion that it protects through suppression of neuronal activity and cerebral energy demand. In this study, we combined PAM-based optical measurement of $CMRO_2$ and functional assays of acutely purified cortical mitochondria in HI-injured awake mouse neonates to investigate the causes of post-HI SEF and hypothermia protection. Our study has three unique features. First, awake-brain imaging avoids the confounding effects of anesthesia on CBF and $CMRO_2$ (*Cao et al., 2017*; *Gao et al., 2017*; *Sciortino et al., 2021*; *Slupe and Kirsch, 2018*). Second, our PAM-based $CMRO_2$ measurement is analogous to optical $CMRO_2$ detection using bedside instruments that may enable non-invasive monitoring of neonatal brain injury (*De Carli et al., 2019*; *Dehaes et al., 2014*; *Ferradal et al., 2017*; *Jain et al., 2014*; *Liu et al., 2014a*). Third, our experiments establish a direct connection between the impacts of HI on $CMRO_2$ and that on mitochondria, which account for the majority of cerebral oxygen metabolism (*Rolfe and Brown, 1997*). In the following, we discuss the implications of our findings on the cause of post-HI SEF, the protective mechanisms of hypothermia in HI, and the potential of optic $CMRO_2$-detection in neonatal care.

### Causes of post-HI SEF: uncoupled OXPHOS and excessive ROS

In normal conditions, $CMRO_2$ remains relatively stable over a range of CBF and brain glucose-uptake changes as the so-called 'uncoupling of CBF and cerebral oxygen metabolism' (*Fox and Raichle, 1986*). The autoregulation of $CMRO_2$, accomplished by alterations of OEF (an index of oxygen diffusion across capillaries) in the opposite direction of CBF fluctuations, maintains a consistent level of mitochondrial respiration, while glycolysis and phosphocreatine-creatine conversion support the transient outbursts of neural activity (*Chen et al., 2023*; *Lin and Powers, 2018*).

As illustrated by our PAM measurements in unilateral CCA-ligation, the IL cortex showed OEF elevation to compensate for a lower CBF than the CL cortex and maintained bilaterally similar $CMRO_2$ values (*Figures 2D and 3D*, and *Supplementary files 1 and 2*). During combined HI, the CL cortex showed up-regulation of both CBF and OEF and produced a higher $CMRO_2$, similar to the responses to moderate hypoxia (10% $O_2$) in uninjured animals (*Figure 1G*). These responses suggest a greater demand for brain energy as adaptation to hypoxia. In contrast, the IL cortex showed a smaller increase of CBF and mild or no changes of OEF in combined HI, leading to a 40–50% reduction in $CMRO_2$ (*Figures 2D and 3D*, and *Supplementary files 1 and 2*). Consistent with our findings, previous studies showed marked reduction of ATP and phosphocreatine, as well as a massive buildup of lactate, in the IL cortex during combined HI, while the CL cortex showed only minimal ATP reduction and modest accumulation of lactate (*Chen et al., 2023*; *Gunn et al., 1997*; *Salford and Siesjö, 1974*). The bilateral disparity of ATP homeostasis-versus-reduction during combined HI may influence the divergent $CMRO_2$ responses post-HI: the CL cortex showed $CMRO_2$ recovery to the pre-HI level because extra ATP output is no longer required to cope with hypoxia, while the IL cortex developed a surge of mitochondrial respiration (and thus $CMRO_2$) presumably to recompense the energy deficit incurred during HI (*Figure 2D and F*).

A persistent increase in oxygen consumption, albeit to a lesser degree, was also observed in post-ischemic myocardium and after spreading depression in rodent cerebral cortex (*Benzi and Lerch, 1992*; *Juhaszova et al., 2004*; *Piilgaard and Lauritzen, 2009*). Moreover, brain-injured infants showed a higher $CMRO_2$ than healthy neonates (*Grant et al., 2009*), suggesting that the rebound of oxygen metabolism is a generic response to transient severe HI. Notably, our PAM measurements showed that the rise of post-HI $CMRO_2$ is primarily driven by the increase of OEF (*Figure 2C* and *Supplementary file 1*), which may be due to reduction of capillary transient time heterogeneity (*Paulson et al., 2010*). The HI-induced energy deficit and tissue hypoxia may also assist oxygen diffusion across capillaries to elevate the OEF (*Chen et al., 2023*; *Salford and Siesjö, 1974*; *Yang et al., 2009*). Finally, post-HI acidosis may also promote mitochondrial respiration (*Jespersen and Østergaard, 2012*; *Khacho et al., 2014*). We suggest that these cellular and vascular factors collectively promote mitochondrial respiration and $CMRO_2$ immediately after HI (*Figure 5A*).

However, the post-HI mitochondrial respiration has features of OXPHOS uncoupling and ROS release, as shown by the rise of oxygen consumption and mitochondrial superoxide despite the reduction of mitochondrial membrane potential needed for ATP-synthesis through F0/F1 ATPase (*Figure 2F–H*). The triggers of OXPHOS uncoupling after HI may include persistent tissue hypoxia (*Kramer and Pearlstein, 1983*; *Kuan et al., 2021*), influx of calcium in the mitochondrial matrix (*Benzi and Lerch, 1992*; *Kristián and Siesjö, 1998*), reverse electron transfer through Complex I (*Kim et al., 2018*), nitric oxide-mediated inhibition of the cytochrome oxidase in Complex IV (*Cooper and Giulivi, 2007*), and ROS-caused disruption of the electron transport chain super-complexes and mitochondrial cristae (*Paradies et al., 2018*), according to the literature and our results (*Figure 5—figure supplement 1*). Importantly, the release of extra ROS during uncoupled OXPHOS injures the mitochondria (*Granger and Kvietys, 2015*; *Murphy, 2009*), as indicated by the reduction of oxygen consumption and mitochondrial membrane potential at 5 hr post-HI, if without hypothermia treatment (*Figure 2F–H*). These results suggest that uncoupled OXPHOS and ROS-mediated injury of mitochondria are important mechanisms of post-HI SEF (*Figure 5A*).

## Protective mechanisms of hypothermia

If the aforementioned hypothesis is correct, therapeutic hypothermia shall attenuate the post-HI $CMRO_2$-surge. Indeed, our data showed that the initiation of mild hypothermia (32 °C) post-HI efficiently clamped the rebounding $CMRO_2$ to the pre-HI level at both hemicortices over the 2 hr PAM-monitoring period (*Figure 3D*). Furthermore, hypothermia blocked the rapid demise of mitochondria at 5 hr post-HI (*Figure 3F–I*) and preserved cortical ATPs at 6 hr post-HI (*Figure 3J*). In other words, neonatal mice recovering under mild hypothermia after HI escaped the initial overshoot of $CMRO_2$ and the subsequent SEF, as predicted by our hypothesis (*Figure 5A*).

In terms of mechanism, our PAM measurements showed that hypothermia mainly blocks the rise of post-HI OEF to constrain $CMRO_2$ (*Figure 3E* and *Supplementary file 2*). This is different from the responses to mild hypothermia (32 °C) in uninjured mice, which showed CBF reduction but negligible changes in OEF (*Figure 1E*). This difference suggests unique effects of hypothermia in the post-HI

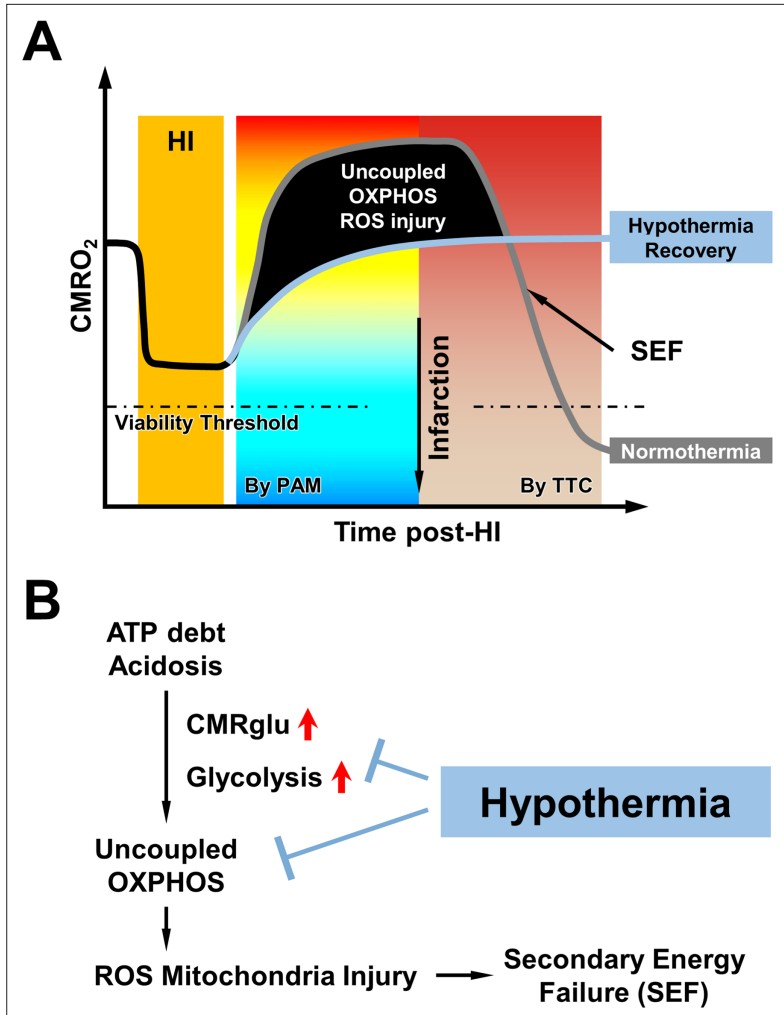

**Figure 5.** Schematic conclusion and open questions. (**A**) Combined hypoxia-ischemia (HI) initially suppresses cerebral oxygen metabolism but sparks a rapid rebound and overshoot of post-HI $CMRO_2$ caused by uncoupled OXPHOS with greater ROS emission, which leads to the demise of mitochondria and the onset of secondary energy failure (SEF). In contrast, recovery at hypothermia attenuates the post-HI surge of mitochondrial respiration and produces more enduring cerebral oxygen metabolism. The threshold of $CMRO_2$ reduction that causes brain infarction in neonates is expected to exist but yet to be established. (**B**) The combination of energy deficit and tissue acidosis accumulated during HI plus increased glucose uptake and glycolysis acutely after HI stimulates uncoupled OPXHOS in the HI-injured neonatal brain, leading to excessive ROS emission and rapid mitochondrial injury. According to the literature (*Berntman et al., 1981*; *Hägerdal et al., 1975*) and our results, hypothermia may interrupt this pathological process by inhibiting glycolysis and the TCA cycle.

The online version of this article includes the following figure supplement(s) for figure 5:

**Figure supplement 1.** Potential mechanisms that contribute to the uncoupling of OXPHOS after HI and the effects of therapeutic hypothermia.

brain. Previous studies showed that mild hypothermia represses glycolysis and the tricarboxylic acid (TCA) cycle at the phosphofructokinase and isocitrate dehydrogenase step, respectively (*Berntman et al., 1981*; *Hägerdal et al., 1975*). Consequently, the buildup of lactate and acidosis and the electron transport chain reactions in OXPHOS are attenuated, which may repress post-HI mitochondrial respiration and prevent SEF and brain injury according to our hypothesis (*Figure 5B*).

Whether hypothermia reduces post-HI brain glucose uptake, however, remains unclear. Although previous studies showed a reverse correlation of cerebral glucose metabolism (CMRglu) and the severity of HIE in neonates at 4–24 days of age (*Thorngren-Jerneck et al., 2001*), the responses of CMRglu in the acute post-HI phase remain unknown. If post-HI CMRglu is similarly up-regulated due

to energy deficit, the influx of glucose and the ensuing glycolysis may stimulate mitochondrial respiration to produce greater ROS injury, while hypothermia typically represses CMRglu and could reduce HI injury in this scenario (*Erecinska et al., 2003*). Future studies are thus warranted to examine the impacts of HI with or without hypothermia treatment on CMRglu in mouse neonates.

## Optical CMRO$_2$ detection in neonatal care

Early detection of brain injury in neonatal care has profound importance for early intervention, treatment optimization, and outcome prognostication. Although MRI is the gold standard for detecting acute brain injury, its application in early diagnosis and HIE therapy is hampered by non-portability, high cost, and susceptibility to motion. In contrast, advances in combined near-infrared spectroscopy (NIRS) and diffuse correlation spectroscopy (DCS) have enabled optical CMRO$_2$ detection as a non-invasive, bedside procedure in neonatal care (*De Carli et al., 2019*; *Dehaes et al., 2014*; *Ferradal et al., 2017*; *Jain et al., 2014*; *Liu et al., 2014a*). This is owing to the relatively thin infant skull compared to that of adults and the separation of anterior and posterior fontanelle sutures, which facilitate the penetration of infrared light to quantify the brain tissue oxygenation. The combination of NIRS-based cerebral oximetry and DCS-based CBF mapping in the same area enables quantitative measurements of the regional CMRO$_2$ index or CMRO$_{2i}$. The NIRS/DCS-measured CMRO$_{2i}$ correlates well with the MRI-measured CMRO$_2$ in neonates and declines during hypothermic cardiopulmonary bypass (*Dehaes et al., 2014*; *Ferradal et al., 2017*; *Jain et al., 2014*). A phase 1 clinical trial (NCT02815618) is underway to determine the safety of NIRS/DCS and the normal CMRO$_{2i}$ in newborn infants (*De Carli et al., 2019*). Notably, our PAM technique is analogous to NIR/DCS, and both methods measure CMRO$_2$ in the cortical layers. Moreover, a past NIR/DCS study suggested increased cerebral oxygen consumption in infants with HIE symptoms (*Grant et al., 2009*), which echoes our findings. Thus, our results may shed insights into the utility of optical CMRO$_2$ detection in neonatal care. Additionally, the spatial heterogeneity in estimated CMRO$_2$ observed in our data may reflect underlying physiological variability, including differences in vascular structure or metabolic demand across cortical regions. Future studies will aim to further validate and interpret these spatial patterns.

In adult ischemic stroke patients, CMRO$_2$ can be used to differentiate the infarct core and the adjacent penumbra area, where brain cells are energy-strained but remain salvageable if local blood oxygen supply is restored promptly (*Lin and Powers, 2018*). The cut-off for 'salvageable penumbra' is usually at <60% reduction of CMRO$_2$ (*Lee et al., 2003*). Our results showed that HI-injured mouse neonates preserved 85% of CMRO$_2$ at 24 hr post-HI if recovered under mild hypothermia (32 °C), but lost ~75% of CMRO$_2$ without this treatment (*Figure 4G*). Further, the severity of CMRO$_2$ reduction at 24 hr correlates with infarction, as shown by the results of TTC-stain immediately after PAM-imaging (*Figure 4B and C*). These results suggest that a minimal level of CMRO$_2$ to prevent infarction may also exist in infants after HI, as in adults after ischemic stroke, despite the fact that the normal CMRO$_2$ in infants is lower than that in adults and may differ between term and preterm neonates (*Altman et al., 1993*; *Liu et al., 2014b*). Since NIRS/DCS-based CMRO$_{2i}$ measurement is a non-invasive procedure, it can be performed repeatedly to detect early brain injury through the trajectory of CMRO$_{2i}$ alterations in neonates with clinical HIE symptoms.

## Limitations in this study

While P10 mice are widely used to model near-term human infants, developmental differences in cellular metabolism and neurovascular coupling may affect the observed outcomes and limit direct clinical translation (*Clancy et al., 2007*; *Mallard and Vexler, 2015*; *Sheldon et al., 2018*). Nevertheless, the P10 model remains a valuable and widely accepted tool for studying neonatal hypoxia-ischemia mechanisms and evaluating therapeutic interventions.

A technical limitation is the absence of direct intracortical temperature measurements during hypothermia; we relied on skull temperature, which may not fully capture temperature dynamics in deeper cortical layers. However, this approach aligns with clinical practice, where intracortical temperature is not typically measured. Future studies could benefit from more precise intracortical assessments. Another limitation of this study is the restricted imaging depth of the PAM technique, which is typically less than 1 mm and therefore does not allow assessment of deeper brain structures such as the basal ganglia. Yet, in experimental models and clinical cases of neonatal hypoxia-ischemia, the cortical

injury tends to be more prominent and functionally significant. As such, our results remain relevant for investigating the pathological mechanisms and therapies of this neonatal brain disorder.

While our study focuses on the acute effects of hypothermia, previous research has shown long-term neuroprotective benefits, including improved white matter development post-injury (*Koo et al., 2017*). These findings highlight the potential of hypothermia for both immediate and extended recovery, warranting further study of long-term outcomes.

Lastly, for awake imaging, the small size of neonatal mice at P10 aids stability during awake PAM imaging, though it limits the feasibility of prior training, which is typically possible in older animals. We observed no signs of distress or pain and did not use stress- or pain-reducing drugs during imaging. However, potential effects of stress or residual pain on CBF and $CMRO_2$ cannot be fully ruled out. Future studies could incorporate more detailed pain assessment and stress-mitigation strategies to further enhance physiological reliability.

## Methods

### Multi-parametric PAM system

As shown in *Figure 1A*, a nanosecond-pulsed laser (wavelength: 532 nm, repetition rate: up to 30 kHz; BX40-2-G, EdgeWave) was used in the PAM system for imaging the neonatal mouse brain. The laser beam firstly passed through a half-wave plate (HWP; WPH05M-532, Thorlabs) and an electro-optical modulator (EOM; 350–80, Conoptics) for precise control of the polarization state of the incident beam. By alteration of the voltage applied to the EOM, the polarization state could be dynamically switched between the vertical direction and the horizontal direction, which allowed a polarizing beam splitter (PBS; PBS121, Thorlabs) to dispatch the laser pulses between two optical paths through either reflection or transmission. In the reflection path, after being partially attenuated by a neutral-density filter (NDF; NDC-50C-2M, Thorlabs), the beam was coupled through a fiber collimator (CFC-11X-A, Thorlabs) into a polarization-maintaining single-mode fiber (PM-SMF; F-SPA, Newport) for stimulated Raman scattering-based wavelength conversion. The output of the PM-SMF was collimated by an identical collimator and purified by a bandpass filter (BPF; FB560-10, Thorlabs) to isolate the 558 nm component. Then, the 558 nm Raman beam generated in the reflection path and the 532 nm beam in the transmission path were combined by a dichroic mirror (DBS; FF538-FDi01, Semrock) and coupled into a single-mode fiber (SMF; P1-460B-FC-2, Thorlabs) through a fiber collimator (CFC-11X-A, Thorlabs), before which ~5% of the combined beam was picked off by a beam sampler (BS; BSF10-A, Thorlabs) and monitored by a high-speed photodiode (PD; FDS100, Thorlabs) to compensate for possible fluctuation in the laser energy. The dual-wavelength laser beam was then delivered to the scanning head, where two identical doublets (DL; AC127-025-A, Thorlabs) were used to map the fiber output into the tissue to be imaged, and a correction lens (CL; LA1207-A, Thorlabs) was used to compensate for the optical aberration at the air-water interface. A ring-shaped ultrasonic transducer (UT; inner diameter: 2.2 mm; outer diameter: 4.0 mm; center frequency: 35 MHz; 6 dB bandwidth: 70%) was used for confocal alignment of the optical excitation and ultrasonic detection. A custom-ized water tank (WT) was used to immerse the ultrasonic transducer for acoustic coupling. As shown in *Figure 1B*, the mouse head was fixed to a customized metal arm piece through a wearable 3-D printed head plate for head-restrained awake-brain imaging. A water-floated treadmill (03170–1008, Blick Art Materials) was used to allow the mouse to move freely with reduced reaction force (*Cao et al., 2017*). A gas inlet and a customized face mask from the syringe were used to deliver the inhalation gas (i.e. normoxia or hypoxia).

### Procedures for PAM imaging

Before imaging of the neonatal mouse brain, the animal was first anesthetized with vaporized isoflurane (EZ-SA800, E-Z Systems) for installation of the head-restraint plastic head plate (*Sciortino et al., 2021*). To manage pain, 0.25% Bupivacaine was administered locally prior to the surgical procedures. Hair on the mouse scalp was removed by a trimmer (9990–1301, Wahl Clipper), and then the scalp was removed with surgical scissors (MDS10030, Medline Industries) to expose the skull. After the clearance of the blood, debris, and remaining hairs, the gel-based cyanoacrylate glue (234790, Loctite) was applied to all edges of the head plate, which was attached to the center of the skull between the Bregma and Lambda. After the applied glue is solidified (~20 min), the frame-worn animal was first

returned to its cage for full recovery from anesthesia, and then carefully moved to the treadmill and secured to the metal arm-piece with two #4–40 screws for awake PAM imaging. The total duration of anesthesia, including preparation and glue solidification, was approximately 20 min. Throughout the frame installation, the body temperature of the mouse was maintained at 37 °C using a homeothermic monitoring system (No. 69020, RWD life science).

After the installation of the head-restraint head plate, a thin layer (~1 mm) of ultrasound gel (Aquasonic CLEAR, Parker Laboratories) was applied to the surface of the skull for effective acoustic coupling. Then, the metal arm piece and the treadmill were carefully raised to bring the gel in gentle contact with the bottom of the water tank, which was filled with temperature-maintained deionized water. After fully recovering from the anesthesia, the neonatal mouse was ready for awake-brain imaging by the multi-parametric PAM system, which was controlled by a field-programmable gate array (PCIe-7841R, National Instruments) through a self-developed LabVIEW program. For the imaging field covering both hemicortices between the Bregma and Lambda of the neonatal mouse (5×3 mm$^2$ as shown in *Figure 1C*, with each hemicortex measuring 2.5×3 mm$^2$), the acquisition time is ~40 min with step sizes set to 0.1 and 10 µm along the *x*- and *y*-direction, respectively. This study was conducted in strict accordance with the Guide for the Care and Use of Laboratory Animals of the National Institutes of Health and the ARRIVE (Animal Research: Reporting of In Vivo Experiments) guidelines. All procedures were approved by the Institutional Animal Care and Use Committees (IACUC) at Washington University in St. Louis (#25–0161) and the University of Virginia (#4209–0218).

## PAM imaging under normothermia or hypothermia

The temperature of the water tank was controlled by a temperature controller (EW-89802–52, Cole-Parmer), which could also be used to effectively regulate the temperature of the underlying mouse skull. Based on our calibration experiment (*Figure 1—figure supplement 1*), the normothermia condition (37 °C) could be achieved with the water temperature set to 39 °C, while for the two hypothermia conditions (32 °C or 29 °C), the water temperature was set to 33 °C or 29 °C, respectively. Throughout all the experiments, the skull temperature was closely monitored by a thin-film temperature sensor (F3132, Omega) attached to the skull. Note that the PAM measurements were performed one hour after each temperature adjustment to ensure equilibrium under the new temperature setting.

## PAM imaging under normoxia or hypoxia

For normoxia, the oxygen concentration in the inhalation gas was 21% (AI M-T, Praxair). For hypoxia, the medical-grade air was mixed with medical-grade nitrogen gas (NI-H, Praxair) through a gas flow-meter mixer (EW-03218–56, Cole-Parmer) to achieve a reduced oxygen concentration of 10%, which was confirmed by a clinical anesthesia monitor (Capnomac Ultima, Datex-Ohmeda). Under both conditions, the flow rate of the inhalation gas was set to 1.5 L/min.

## PAM imaging of the Vannucci HI model

In the Vannucci HI study, the first PAM image set was acquired at 37 °C after the unilateral CCA ligation. Then, the second image set was acquired at 37 °C during the 1-hr HI challenge. Subsequently, three sequential image sets were acquired up to 120 min (i.e. 0–40, 40–80, and 80–120 min) post-HI, either under normothermia (37 °C) or hypothermia (32 °C) by regulating the temperature of the water tank.

## Quantification of cerebral hemodynamics and oxygen metabolism by PAM

Our PAM technique enables simultaneous quantification of multiple microvascular parameters, including $C_{Hb}$, $sO_2$, and blood flow by the statistical, spectroscopic, and correlation analyses of the acquired A-line signals, respectively (*Cao et al., 2017*). In this process, data analysis was performed without knowledge of the temperature treatment group, and no animals were excluded from the analysis unless death occurred prior to completion of the experiment. With our self-developed MATLAB-based vessel segmentation algorithm (*Sun et al., 2020*), these hemodynamic parameters could be extracted at the single-microvessel level. Briefly, this process involves generating a vascular map using signal amplitude obtained from the Hilbert transformation, selecting a region slightly larger than the vessel of interest, and applying Otsu's thresholding method to remove background pixels. Isolated or

spurious boundary fragments are then removed to improve boundary smoothness. The customized MATLAB code used for vessel segmentation in this study is publicly available at https://github.com/HuLab-WashU/PAM-Otsu-Segmentation (*Sun, 2025*). Then, CBF, OEF, and CMRO$_2$ could be obtained by the following formulas:

$$CBF = \pi v d^2/8$$

$$OEF = \left(s_aO_2 - s_vO_2\right)/s_aO_2$$

$$CMRO_2 = \xi \times C_{Hb} \times s_aO_2 \times OEF \times CBF$$

where $d$ is the vascular diameter, $v$ is the peak flow speed along the vascular axis, $s_aO_2$ and $s_vO_2$ are the $sO_2$ values of the feeding arteries and draining veins, respectively (*Cao et al., 2017*), and $\xi$ is the oxygen binding capacity of hemoglobin (0.014 L O$_2$ per gram hemoglobin).

## Neonatal cerebral HI and hypothermia treatment

The Vannucci model of neonatal HI with and without hypothermia treatment was performed in P10 C57BL/6 mice as described (*Chen et al., 2021*; *Kuan et al., 2021*; *Yang et al., 2009*). P10 mice were chosen for our experiments as they are widely used to model near-term infants in humans. At this developmental stage, the brain maturation in mice closely parallels that of near-term infants, making them an appropriate model for studying neonatal brain injury and therapeutic interventions (*Clancy et al., 2007*; *Mallard and Vexler, 2015*; *Sheldon et al., 2018*). Briefly, P10 mice of both sexes anesthetized with 2% isoflurane were subjected to the right CCA-ligation. To manage pain, 0.25% Bupivacaine was administered locally prior to the surgical procedures, which took less than 10 min. After a recovery period for 1 hr, awake mice were exposed to 10% O$_2$ for 40 min in a hypoxic chamber at 37 °C. The mice were recovered to normoxia (21% O$_2$) condition, and then randomly divided into normothermia or hypothermia groups with the chambers submerged in a 37 °C or 32 °C water bath, separately for a total four-hour treatment. The body temperature of mice was monitored at 37 °C or 32 °C using a rectal probe (BAT-12 microprobe, Physitemp) during the treatment. Here, sample sizes were determined based on preliminary results and prior experience with similar experimental paradigms.

## Measurement of brain infarction

Detection of brain infarction was performed by TTC staining (Sigma-Aldrich) at 24 hr post-HI as previously described (*Kuan et al., 2021*; *Yang et al., 2009*). The fresh brains were collected after cold-PBS transcardial perfusion. The whole brain (*Figure 4C*) or 1 mm-thick brain sections (*Figure 4D*) were incubated with 2% TTC solution, and the TTC-unstained volume in all sections was quantified using the NIH ImageJ software.

## HR-MAS NMR study

The ex-vivo NMR experiments and analyses of mice brain tissues were performed as previously reported (*Chen et al., 2021*). Briefly, a 1.5 mm punch (~10 mg) was taken from snap-frozen brain tissue and loaded into a sample rotor (4 mm ZrO$_2$, Bruker Instruments), with 4 µL of deuterium oxide containing 100 mM sodium trimethylsilylpropionate-d4 (TSP, Sigma-Aldrich) added to obtain a frequency-lock signal and serve as an internal standard for chemical shift. HR-MAS NMR experiments were then carried out on an NMR spectrometer (AVANCE 400 WB, Bruker) with a dedicated 4 mm HR-MAS probe. Spinning rate of samples was set to 2500 kHz (±2 Hz) at 4 °C. A T2-weighted, water-suppressed Carr-Purcell-Meiboom-Gill pulse sequence was used to acquire the data. The $^1$H-NMR spectra were recorded using key parameters as follows: repetition time of 2.0 s, spectral width of 4.8 kHz, 32 K data points, and 256 transients. The presence and concentrations of selected metabolites in brain tissue samples were determined based on their chemical shifts and corresponding integrals using the Electronic REference To access In vivo Concentration (ERETIC) reference method (Bruker), with 10 mM TSP as the external standard.

## ATP detection

The brain ATP level was measured through an ATP Assay Kit (Catalog No. ab83355, Abcam) according to the manufacturer's instructions. Briefly, mouse brains were harvested, washed with 1 X PBS, and

then resuspended in 100 µL of ATP assay buffer. Cells were homogenized and then centrifuged at 4 °C at 12,000 × $g$ to remove the insoluble material. The supernatants were collected and incubated with the ATP probe. Absorbance was detected at 580 nm using a microplate reader (SpectraMax M3 Microplate Reader, Molecular Devices).

## Assessment of isolated brain mitochondrial function

Mitochondria were isolated from the brain tissues using Percoll density gradient centrifugation as described before (*Caspersen et al., 2008*; *Niatsetskaya et al., 2012*). Mitochondrial respiration was measured by a Clark-type electrode (Oxytherm, Hansatech). Briefly, 0.05 mg mitochondria protein was added into 0.5 mL respiration buffer at pH 7.2, which consists of 200 mM sucrose, 25 mM KCl, 2 mM $K_2HPO_4$, 5 mM HEPES, 5 mM $MgCl_2$, 0.2 mg/mL of BSA, 30 µM $P^1$, $P^5$-di(adenosine 5')-pentaphosphate (Ap5A), 10 mM glutamate, and 5 mM malate at 32 °C, with the addition of 5 mM succinate to trigger respiration. As shown in *Figure 2—figure supplement 1*, to initiate State 3 phosphorylating respiration, 100 nmol of adenosine diphosphate (ADP) was added to the mitochondrial suspension.

The rate of $O_2$ consumption was presented in nmol $O_2$/mg mitochondrial protein/min. The respiratory control ratio was calculated as the ratio of the State 3 respiration rate versus the State 4 resting respiration rate recorded after the phosphorylation of ADP has been finished. 35 nM 2'- 4' Dinitrophenol (DNP) was used to initiate uncoupled respiration (*Kim et al., 2018*).

The rate of $H_2O_2$ emission from isolated mitochondria was measured using a fluorescence assay by a fluorescence spectrophotometer (F-7000, Hitachi High Technologies America, Inc), which was set at 555 nm excitation and 581 nm emission as previous study (*Starkov and Fiskum, 2003*). Briefly, 0.05 mg mitochondria were added into 1 mL respiration buffer and supplemented with 5 mM succinate, 10 µM Amplex Ultrared (Catalog No. A22180, Invitrogen), and 4 U/mL horseradish peroxidase (HRP). After recording the fluorescence for 400 s, samples were added with 1 µM rotenone, then added with 1 µg/mL antimycin A after another 200 s. The calibration curve was calculated by sequential additions of known amounts of $H_2O_2$, with the cuvette containing the respiration buffer, Amplex Ultrared, and HRP.

Superoxide was monitored with MitoSOX Red (Catalog No. M36008, Invitrogen). Briefly, 5 µM MitoSOX Red was added to the cuvettes and the fluorescence was measured at 510 nm excitation and 579 nm emission wavelengths. Time-dependent monitoring of the MitoSOX Red-related fluorescence was done using a fluorescence spectrophotometer (F-7000, Hitachi High Technologies America, Inc) at 25 °C. Since a saturation of kinetics established after 25 min of the recording, the initial linear increase rates of the MitoSOX Red-related fluorescence were used for quantification.

For membrane potential analysis, the rhodamine 123 (RH-123; Sigma-Aldrich) quenching technique was conducted. 0.05 mg isolated mitochondria were incubated with 0.3 mM RH-123 and analyzed by a fluorescence spectrophotometer (F-7000, Hitachi High Technologies America, Inc), using excitation and emission wavelengths of 503 nm and 527 nm, respectively. During the measurements, the reaction buffer containing mitochondria was continuously stirred. Baseline fluorescence was recorded for 300 s followed by the sequential addition of oxidizable substrates (7 mM succinate or 7 mM malate and glutamate) and 50 mM carbonyl cyanide m-chlorophenyl hydrazone (CCCP).

## Video-EEG monitoring

Video-EEG monitoring of mouse neonates was performed as described (*Burnsed et al., 2019*). Briefly, nine-day-old C57BL/6 mice had unipolar insulated stainless steel depth electrodes (bare diameter: 0.005 inch, coated: 0.008 inch; A-M Systems, Sequim) stereotactically implanted in the bilateral parietal cortices (−1.2 dorsoventral (DV),±0.5 mediolateral (ML), and −1.0 deep (D) mm), bilateral CA1 region of the hippocampus (−3.5 DV,±2.0 ML, and −1.75 D mm), and a reference electrode in the cerebellum. Following recovery, mice were exposed to HI on P10. A unity gain impedance matching head stage (TLC2274 Quad Low-Noise Rail-to-Rail Operational Amplifier, Texas Instruments) was used for recordings, which began 1 hr prior to CCA ligation and continued through 2 hr after re-oxygenation post-ligation. Mice were then returned to the dam.

## Statistical analysis

One-way ANOVA was used to compare the vascular and oxygen-metabolic changes in the normothermia *vs.* hypothermia experiment (*Figure 1E*), and to assess mitochondrial function in the UN,

CL, and IL cortices (*Figure 2F–H*). A Student t-test was used to compare the hemodynamic and oxygen-metabolic changes in the normoxia *vs.* hypoxia experiment (*Figure 1G*), and the infarct volume of normothermia or hypothermia (*Figure 4D*). Two-way ANOVA was used to compare the oxygen-metabolic changes of the CL and IL cortices in the Vannucci HI model under normothermic (*Figure 2B–D*) or hypothermic (*Figure 3B–E*) condition, the mitochondrial function (*Figure 3F–I*) and ATP level (*Figure 3J*) of CL or IL cortices under hypothermia, and the hemodynamic and oxygen-metabolic changes 24 hr post-HI (*Figure 4E–I*). All data were presented in the form of mean ± standard deviation of the mean. In all statistical analyses, $p < 0.05$ was considered significant.

## Acknowledgements

We thank Dr. Vadim Ten for demonstrating the mitochondria isolation and analysis methods. This work was supported by the National Science and Technology Council of Taiwan MOST 111–2320-B-110–003-MY2, NSTC 113–2320-B-110–010 (to Y-YS), NIH grants NS125788, NS127392, and HD109025 (to C-YK), NS099261 and NS120481 (to SH), and NS125677 (to C-YK and SH), and NSF CAREER 202988 (to SH).

## Additional information

### Funding

| Funder | Grant reference number | Author |
| --- | --- | --- |
| National Science and Technology Council | MOST 111-2320-B-110-003-MY2 | Yu-Yo Sun |
| National Institutes of Health | NS125788 | Chia-Yi Kuan |
| National Institutes of Health | NS099261 | Song Hu |
| National Institutes of Health | NS125677 | Chia-Yi Kuan Song Hu |
| National Science Foundation | NSF CAREER 202988 | Song Hu |
| National Science and Technology Council | NSTC 113-2320-B110-010 | Yu-Yo Sun |
| National Institutes of Health | NS127392 | Chia-Yi Kuan |
| National Institutes of Health | HD109025 | Chia-Yi Kuan |
| National Institutes of Health | NS120481 | Song Hu |

The funders had no role in study design, data collection and interpretation, or the decision to submit the work for publication.

### Author contributions

Naidi Sun, Data curation, Software, Formal analysis, Validation, Investigation, Methodology, Writing – original draft; Yu-Yo Sun, Data curation, Formal analysis, Funding acquisition, Validation, Investigation, Methodology, Writing – original draft; Rui Cao, Hong-Ru Chen, Yiming Wang, Formal analysis, Methodology; Elizabeth Fugate, Marchelle R Smucker, Yi-Min Kuo, Methodology; Ellen P Grant, Diana M Lindquist, Investigation, Visualization, Methodology; Chia-Yi Kuan, Song Hu, Conceptualization, Resources, Data curation, Supervision, Funding acquisition, Validation, Investigation, Visualization, Project administration, Writing – review and editing

### Author ORCIDs

Naidi Sun ⓘ https://orcid.org/0000-0002-2054-0392

Yu-Yo Sun http://orcid.org/0000-0002-6608-1347
Yi-Min Kuo https://orcid.org/0000-0002-7949-6664
Song Hu https://orcid.org/0000-0002-5760-8012

### Ethics

This study was conducted in strict accordance with the Guide for the Care and Use of Laboratory Animals of the National Institutes of Health and the ARRIVE (Animal Research: Reporting of In Vivo Experiments) guidelines. All procedures were approved by the Institutional Animal Care and Use Committees (IACUC) at Washington University in St. Louis (#25-0161) and the University of Virginia (#4209-0218).

Reviewer #1 (Public review): https://doi.org/10.7554/eLife.100129.3.sa1
Reviewer #3 (Public review): https://doi.org/10.7554/eLife.100129.3.sa2
Author response https://doi.org/10.7554/eLife.100129.3.sa3

## Additional files

### Supplementary files

Supplementary file 1. CBF, OEF, and $CMRO_2$ values measured at 37 °C in the contralateral (green) and ipsilateral (red) cortex in the indicated period that correspond to those illustrated in *Figure 2A*. Data are shown in mean ± SD and compared to the "CCA-ligation" on its own side, unless indicated with a bracket. The *p*-values were determined by two-way ANOVA (n=5; *: $P<0.05$; **: $P<0.01$).

Supplementary file 2. CBF, OEF, and $CMRO_2$ values with post-HI hypothermia treatment, measured at 32 °C in the contralateral (green) and ipsilateral (red) cortex in the indicated period that correspond to those illustrated in *Figure 3A*. Data are shown in mean ± SD and compared to the "CCA-ligation" on its own side, unless indicated with a bracket. The *p*-values were determined by two-way ANOVA (n=5; ***: $P<0.001$; ****: $P<0.0001$).

MDAR checklist

### Data availability

All data is included in the Source data files.

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
