## [Editor Report · eLife Assessment]

This is an **important** study that utilized in vivo optical measurements of the cortical metabolic rate of O2 and blood flow, as well as measurements in isolated mitochondria to assess the uncoupling of the oxidative phosphorylation due to hypoxia-ischemia injury of the neonatal brain, and effects of the hypothermia treatment. The combination of state-of-the-art optical measurements, mitochondrial assays, and the use of various control experiments provides **convincing** evidence for the derived conclusions. This work will be of interest to those in the mitochrondrial metabolomics, brain injury and hypoxia fields.

---

## [Referee Report · Reviewer #1 (Public review)]

Summary:

This manuscript addresses the important problem of the uncoupling of oxidative phosphorylation due to hypoxia-ischemia injury in the neonatal brain and provides insight into the neuroprotective mechanisms of hypothermia treatment.

Strengths:

The authors used a combination of in vivo imaging of awake P10 mice and experiments on isolated mitochondria to assess various key parameters of brain metabolism during hypoxia-ischemia with and without hypothermia treatment. This unique approach resulted in a comprehensive data set that provides solid evidence to support the derived conclusions.

Weaknesses:

Several potential weaknesses were identified in the original submission, which the authors subsequently addressed in the revised manuscript. Here is the brief list of the questions:

(1) Is it possible that the observed relatively low baseline OEF and trends of increased OEF and CBF over several hours after the imaging start were partially due to slow recovery from anesthesia?

(2) What was the pain management, and is there a possibility that some of the observations were influenced by the pain-reducing drugs or their absence?

(3) Were P10 mice significantly stressed during imaging in the awake state because they didn't have head-restraint habituation training?

(4) Considering high metabolism and blood flow in the cortex, it could be potentially challenging to predict cortical temperature based on the skull temperature, particularly in the deeper part of the cortex.

(5) The map of estimated CMRO2 looks quite heterogeneous across the brain surface. Could this be partially resulting from the measurement artefact?

(6) It would be beneficial to provide more detailed justification for using P10 mice in the experiments.

---

## [Referee Report · Reviewer #3 (Public review)]

Sun et al. present a comprehensive study using a novel photoacoustic microscopy setup and mitochondrial analysis to investigate the impact of hypoxia-ischemia (HI) on brain metabolism and the protective role of therapeutic hypothermia. The authors elegantly demonstrate three connected findings: (1) HI initially suppresses brain metabolism, (2) subsequently triggers a metabolic surge linked to oxidative phosphorylation uncoupling and brain damage, and (3) therapeutic hypothermia mitigates HI-induced damage by blocking this surge and reducing mitochondrial stress.

The study's design and execution are great, with a clear presentation of results and methods. Data is nicely presented, and methodological details are thorough.

However, a minor concern is the extensive use of abbreviations, which can hinder readability. As all the abbreviations are introduced in the text, their overuse may render the text hard to read to non-specialist audiences. Additionally, sharing the custom Matlab and other software scripts online, particularly those used for blood vessel segmentation, would be a valuable resource for the scientific community. In addition, while the study focuses on the short-term effects of HI, exploring the long-term consequences and definitively elucidating HI's impact on mitochondria would further strengthen the manuscript's impact.

Despite these minor points, this manuscript is very interesting.

Comments on revisions:

All addressed.

---

## [Author Response]

The following is the authors’ response to the original reviews.

**Reviewer #1 (Public review)**
(1) This manuscript addresses an important problem of the uncoupling of oxidative phosphorylation due to hypoxia-ischemia injury of the neonatal brain and provides insight into the neuroprotective mechanisms of hypothermia treatment.The authors used a combination of in vivo imaging of awake P10 mice and experiments on isolated mitochondria to assess various key parameters of the brain metabolism during hypoxia-ischemia with and without hypothermia treatment. This unique approach resulted in a comprehensive data set that provides solid evidence for the derived conclusions

We thank the reviewer for the positive feedback.

(2) The experiments were performed acutely on the same day when the surgery was performed. There is a possibility that the physiology of mice at the time of imaging was still affected by the previously applied anesthesia. This is particularly of concern since the duration of anesthesia was relatively long. Is it possible that the observed relatively low baseline OEF (~20%) and trends of increased OEF and CBF over several hours after the imaging start were partially due to slow recovery from prolonged anesthesia? The potential effects of long exposure to anesthesia before imaging experiments were not discussed.

We thank the reviewer for this important comment and for pointing out the potential influence of anesthesia on the physiological state of the animals. We apologize for any confusion. To clarify, all PAM imaging experiments were conducted in awake animals. Isoflurane anesthesia was used only during two brief surgical procedures: (1) the installation of the head-restraint plastic head plate and (2) the right common carotid artery (CCA) ligation. Each anesthesia session lasted less than 20 minutes.

We have revised the Methods section to provide additional details:

For the subsection Procedures for PAM Imaging on page 17, we clarified the sequence of procedures during the head plate installation, as well as the corresponding anesthesia duration:

“After the applied glue was solidified (~20 min), the animal was first returned to its cage for full recovery from anesthesia, and then carefully moved to the treadmill and secured to the metal arm-piece with two #4–40 screws for awake PAM imaging. The total duration of anesthesia, including preparation and glue solidification, was approximately 20 minutes.”

For the subsection Neonatal Cerebral HI and Hypothermia Treatment on page 19, we also clarified the CCA ligation procedure:

“Briefly, P10 mice of both sexes anesthetized with 2% isoflurane were subjected to the right CCA-ligation. To manage pain, 0.25% Bupivacaine was administered locally prior to the surgical procedures, which took less than 10 minutes. After a recovery period for one hour, awake mice were exposed to 10% O_2_ for 40 minutes in a hypoxic chamber at 37 °C.”

Regarding the reviewer’s concern about the observed trends in OEF and CBF, we agree that residual effects of anesthesia could, in principle, influence physiological parameters. However, we believe this is unlikely in this study for the following reasons. First, all imaging was conducted in awake animals after a clearly defined recovery period. Second, the trend of increasing OEF and CBF over time was consistent across animals and aligned with expected physiological responses following hypoxic-ischemic injury. In particular, the relatively low baseline OEF (0.21 at 37°C) is consistent with our previous study (0.25; (Cao et al., 2018)). The gradual increase in CBF and OEF reflects metabolic compensation and reperfusion following hypoxia-ischemia, as previously described (Lin and Powers, 2018). Therefore, we believe the observed changes are of physiological origin rather than anesthesia-related artifacts.

(3) The Methods Section does not provide information about drugs administered to reduce the pain. If pain was not managed, mice could be experiencing significant pain during experiments in the awake state after the surgery. Since the imaging sessions were long (my impression based on information from the manuscript is that imaging sessions were ~4 hours long or even longer), the level of pain was also likely to change during the experiments. It was not discussed how significant and potentially evolving pain during imaging sessions could have affected the measurements (e.g., blood flow and CMRO_2_). If mice received pain management during experiments, then it was not discussed if there are known effects of used drugs on CBF, CMRO_2_, and lesion size after 24 hr.

We thank the reviewer for this valuable comment regarding pain management. We confirm that local analgesia was administered to all animals prior to surgical procedures. Specifically, 0.25% Bupivacaine was applied locally before both the head-restraint plate installation and the CCA ligation. These details have now been clarified in the Methods section:

For the subsection Procedures for PAM Imaging on page 16, we added:

“To manage pain, 0.25% Bupivacaine was administered locally prior to the surgical procedures.”

For the subsection Neonatal Cerebral HI and Hypothermia Treatment on page 18, we added:

“To manage pain, 0.25% Bupivacaine was administered locally prior to the surgical procedures, which took less than 10 minutes.”

To our knowledge, Bupivacaine has minimal systemic effects at the dose used and is unlikely to significantly alter CBF, CMRO_2_, or lesion development (Greenberg et al., 1998). No other analgesics (e.g., NSAIDs or opioids) were administered unless distress symptoms were observed—which did not occur in this study.

Additionally, although imaging sessions were extended (up to 2 hours), animals remained calm and showed no signs of pain or distress during or after the procedures. Throughout the experimental period (up to 24 hours post-surgery), animals were monitored for signs of discomfort (e.g., abnormal activity, breathing, or weight gain), but no additional analgesia was required. The neonatal HI procedures are considered minimally invasive, and based on our protocol and prior experience, local Bupivacaine provides effective analgesia during and after the brief surgeries. We have added a corresponding note in the Discussion section (newly added subsection: Limitations in this study, the last paragraph) on page 15:

“We observed no signs of distress or pain and did not use stress- or pain-reducing drugs during imaging. However, potential effects of stress or residual pain on CBF and CMRO_2_ cannot be fully ruled out. Future studies could incorporate more detailed pain assessment and stress-mitigation strategies to further enhance physiological reliability.”

(4) Animals were imaged in the awake state, but they were not previously trained for the imaging procedure with head restraint. Did animals receive any drugs to reduce stress? Our experience with well-trained young-adult as well as old mice is that they can typically endure 2 and sometimes up to 3 hours of head-restrained awake imaging with intermittent breaks for receiving the rewards before showing signs of anxiety. We do not have experience with imaging P10 mice in the awake state. Is it possible that P10 mice were significantly stressed during imaging and that their stress level changed during the imaging session? This concern about the potential effects of stress on the various measured parameters was not discussed.

We thank the reviewer for this important comment regarding the potential effects of stress during awake imaging. The neonatal mice used in our study were P10, a stage at which animals are still physiologically immature and relatively inactive. Due to their small size and limited mobility, these animals did not struggle or show signs of distress during the imaging sessions. All animals remained calm and stable throughout the procedure, and no stress-reducing drugs were administered.

We agree that, unlike older animals, P10 mice are not amenable to prior behavioral training. However, their underdeveloped motor activity and natural docility at this stage allowed for stable head-restrained imaging without inducing overt stress responses. Although no behavioral signs of stress were observed, we acknowledge that subtle physiological effects cannot be entirely excluded. We have added a brief discussion in the Discussion section (newly added subsection: Limitations in this study, the last paragraph) on page 15:

“Lastly, for awake imaging, the small size of neonatal mice at P10 aids stability during awake PAM imaging, though it limits the feasibility of prior training, which is typically possible in older animals.”

(5) The temperature of the skull was measured during the hypothermia experiment by lowering the water temperature in the water bath above the animal's head. Considering high metabolism and blood flow in the cortex, it could be challenging to predict cortical temperature based on the skull temperature, particularly in the deeper part of the cortex.

We thank the reviewer for this helpful comment and for highlighting an important technical consideration. We acknowledge that we did not directly measure intracortical tissue temperature during the hypothermia experiments. While we recognize that relying on skull temperature may have limitations—particularly in reflecting temperature changes in deeper cortical regions—this approach is consistent with clinical practice, where intracortical temperature is typically not measured. Moreover, prior studies have shown that skull or brain surface temperature generally reflects cortical thermal dynamics to a reasonable extent under controlled conditions (Kiyatkin, 2007). We have added the following note in the Discussion section (newly added subsection: Limitations in this study, the 2^nd^ paragraph) on page 14:

“A technical limitation is the absence of direct intracortical temperature measurements during hypothermia; we relied on skull temperature, which may not fully capture temperature dynamics in deeper cortical layers. However, this approach aligns with clinical practice, where intracortical temperature is not typically measured. Future studies could benefit from more precise intracortical assessments.”

(6) The map of estimated CMRO_2_ (Fig. 4B) looks very heterogeneous across the brain surface. Is it a coincidence that the highest CMRO_2_ is observed within the central part of the field of view? Is there previous evidence that CMRO_2_ in these parts of the mouse cortex could vary a few folds over a 1-2 mm distance?

We appreciate the reviewer’s insightful observation regarding the spatial heterogeneity observed in the estimated CMRO_2_ map (Fig. 4B). This heterogeneity is not a result of scanning bias, as uniform contour scanning was performed across the entire field of view. The higher CMRO_2_ values observed in the central region are unlikely to be artifacts and more likely reflect underlying physiological variability.

Our CMRO_2_ estimation is based on an algorithm we previously developed and validated in other tissues. Specifically, we have successfully applied this algorithm to assess oxygen metabolism in the mouse kidney (Sun et al., 2021) and to monitor vascular adaptation and tissue oxygen metabolism during cutaneous wound healing (Sun et al., 2022). These studies demonstrated the algorithm's capability to capture spatial variations in oxygen metabolism. Although the current application to the brain is novel, the algorithm has been validated in controlled experimental settings and shown to produce consistent results. We acknowledge that the observed range of CMRO_2_ appears relatively broad across a 1–2 mm distance; however, such heterogeneity may arise from local differences in vascular density, metabolic demand, or tissue oxygenation — all of which can vary across cortical regions, even within small spatial scales. We have added a brief note in the Discussion (Subsection: Optical CMRO_2_ detection in neonatal care) on page 13 to acknowledge this point:

“Additionally, the spatial heterogeneity in estimated CMRO_2_ observed in our data may reflect underlying physiological variability, including differences in vascular structure or metabolic demand across cortical regions. Future studies will aim to further validate and interpret these spatial patterns.”

(7) The justification for using P10 mice in the experiments has not been well presented in the manuscript.

We thank the reviewer for pointing out the need to clarify our choice of developmental stage. We chose P10 mice for our hypoxia-ischemia injury model because this stage is widely recognized as developmentally comparable to human term infants in terms of brain maturation. This approach has been validated by several previous studies (Clancy et al., 2007; Mallard and Vexler, 2015; Sheldon et al., 2018). We have added the following clarification to the Methods section (Subsection: Neonatal Cerebral HI and Hypothermia Treatment) on page 18:

“P10 mice were chosen for our experiments as they are widely used to model near-term infants in humans. At this developmental stage, the brain maturation in mice closely parallels that of near-term infants, making them an appropriate model for studying neonatal brain injury and therapeutic interventions (Clancy et al., 2007; Mallard and Vexler, 2015; Sheldon et al., 2018).”

(8) It was not discussed how the observations made in this manuscript could be affected by the potential discrepancy between the developmental stages of P10 mice and human babies regarding cellular metabolism and neurovascular coupling.

We thank the reviewer for raising this important point regarding developmental differences between P10 mice and human infants. We have discussed this issue by adding the following statement to the Discussion section (newly added subsection: Limitations in this study, the 1^st^ paragraph) on page 15, where we summarize the overall study design and model selection:

“While P10 mice are widely used to model near-term human infants, developmental differences in cellular metabolism and neurovascular coupling may affect the observed outcomes and limit direct clinical translation (Clancy et al., 2007; Mallard and Vexler, 2015; Sheldon et al., 2018). Nevertheless, the P10 model remains a valuable and widely accepted tool for studying neonatal hypoxia-ischemia mechanisms and evaluating therapeutic interventions.”

(9) Regarding the brain temperature measurements, the authors should use a new cohort of mice, implant the miniature thermocouples 1 mm, 0.5 mm, and immediately below the skull in different mice, and verify the temperature in the brain cortex under conditions applied in the experiments. The same approach could be applied to a few mice undergoing 4-hr-long hypothermia treatment in a chamber, which will provide information about the brain temperature that resulted in observed protection from the injury.

We thank the reviewer for this helpful recommendation. We fully agree that direct intracortical temperature measurement would provide more accurate insight into thermal dynamics during hypothermia treatment. However, the primary aim of this study was not to characterize the precise intracortical temperature response under hypothermic conditions, but rather to examine the effects of hypothermia on CMRO_2_ and mitochondrial function. Due to the substantial time and resources required to perform direct intracortical temperature monitoring—and considering the technical focus of the current work—we respectfully suggest reserving such investigations for a future study specifically focused on thermal dynamics in hypoxia-ischemia models.

We have acknowledged this limitation in the subsection Limitations in this study of the Discussion on page 15, noting that skull temperature was used as an approximation of brain temperature and that this approach is consistent with clinical practice, where intracortical temperature is typically not measured. We also note that future studies may benefit from more precise assessments using intracortical probes.

(10) The mean values presented in Fig. 4G are much lower than the peak values in the 2D panels and potentially were calculated as the average values over the entire field of view. Please provide more details on how CMRO_2_ was estimated and if the validity of the measurements is expected across the entire field of view. If there are parts of the field of view where the estimation of CMRO_2_ is more reliable for technical reasons, maybe one way to compute the mean values is to restrict the usable data to the more centralized part of the field of view.

We thank the reviewer for this thoughtful comment. We confirm that CMRO_2_ values shown in Figure 4G were calculated as spatial averages over the entire field of view (FOV; ~5 × 3 mm^2^) encompassing both hemicortices, as shown in Figure 1C. Regarding the observed CMRO_2_ values, The apparent difference likely reflects a comparison between two different post-HI time points. Specifically, the ~0.5 value shown for the 37°C ipsilateral group in Figure 4G reflects the average CMRO_2_ measured 24 hours after HI, while the ~1.5 value in Figure 2D (red line) corresponds to CMRO_2_ during the early 0–2 hour post-HI period. The temporal difference accounts for the apparent discrepancy in magnitude. We understand the importance of consistency across the field of view and have clarified this point in the subsection Procedures for PAM Imaging in the Methods on page 17 “For the imaging field covering both hemicortices between the Bregma and Lambda of the neonatal mouse (5 × 3 mm^2^ as shown in Figure 1C, with each hemicortex measuring 2.5 × 3 mm^2^)”, as well as in the Figure 4 legend on page 34 “Correlation of CMRO_2_ and post-HI brain infarction in mouse neonates at 24 hours”.

In our model and setup, CMRO_2_ estimation is spatially robust across the FOV under standard imaging conditions. We recognize, however, that certain peripheral regions may be more prone to signal attenuation. Future refinement of region selection could further improve spatial averaging strategies. For the current study, full-FOV averaging was used consistently across all groups to maintain comparability.

(11) Minor: Results presented in Supplementary Tables have too many significant digits.

Thank you for the helpful suggestion. We have revised Supplementary Tables S1 and S2 to reduce the number of significant digits and improve clarity.

**Reviewer #2 (Public review)**
(1) In this study, authors have hypothesized that mitochondrial injury in HIE is caused by OXPHOS-uncoupling, which is the cause of secondary energy failure in HI. In addition, therapeutic hypothermia rescues secondary energy failure. The methodologies used are state-of-the art and include PAM technique in live animal, bioenergetic studies in the isolated mitochondria, and others.The study is comprehensive and impressive. The article is well written and statistical analyses are appropriate.

We thank the reviewer for the positive feedback.

(2) The manuscript does not discuss the limitation of this animal model study in view of the clinical scenario of neonatal hypoxia-ischemia.

We thank the reviewer for this valuable feedback. In response, we have added a dedicated “Limitations in this study” subsection in the Discussion, where we address the potential limitations of this animal model in the context of the clinical scenario of neonatal hypoxia-ischemia in the first paragraph on page 14, including the developmental differences between P10 mice and human infants.

(3) I see many studies on Pubmed on bioenergetics and HI. Hence, it is unclear what is novel and what is known.

We thank the reviewer for this important comment regarding the novelty of our study in the context of existing research on bioenergetics and hypoxia-ischemia (HI). To better clarify the novel aspects of our work, we have highlighted the relevant content in the Abstract (page 4) and Introduction (page 5). Specifically, while many studies have explored HI-related bioenergetic dysfunction, the mechanisms by which therapeutic hypothermia modulates CMRO_2_ and mitochondrial function post-HI remain poorly understood.

Abstract on page 4: “However, it is unclear how post-HI hypothermia helps to restore the balance, as cooling reduces CMRO_2_. Also, how transient HI leads to secondary energy failure (SEF) in neonatal brains remains elusive. Using photoacoustic microscopy, we examined the effects of HI on CMRO_2_ in awake 10-day-old mice, supplemented by bioenergetic analysis of purified cortical mitochondria.”

Introduction on page 5: “The use of awake mouse neonates avoided the confounding effects of anesthesia on CBF and CMRO_2_ (Cao et al., 2017; Gao et al., 2017; Sciortino et al., 2021; Slupe and Kirsch, 2018). In addition, we measured the oxygen consumption rate (OCR), reactive oxygen species (ROS), and the membrane potential of mitochondria that were immediately purified from the same cortical area imaged by PAM. This dual-modal analysis enabled a direct comparison of cerebral oxygen metabolism and cortical mitochondrial respiration in the same animal. Moreover, we compared the effects of therapeutic hypothermia on oxygen metabolism and mitochondrial respiration, and correlated the extent of CMRO_2_-reduction with the severity of infarction at 24 hours after HI. Our results suggest that blocking HI-induced OXPHOS-uncoupling is an acute effect of hypothermia and that optical detection of CMRO_2_ may have clinical applications in HIE.”

In this study, we propose that uncoupled oxidative phosphorylation (OXPHOS) underlies the secondary energy failure observed after HI, and we demonstrate that hypothermia suppresses this pathological CMRO_2_ surge, thereby protecting mitochondrial integrity and preventing injury. Additionally, our use of photoacoustic microscopy (PAM) in awake neonatal mice represents a novel, non-invasive approach to track cerebral oxygen metabolism, with potential clinical relevance for guiding hypothermia therapy.

(4) What are the limitations of ex-vivo mitochondrial studies?

We thank the reviewer for this insightful comment. We acknowledge that ex-vivo mitochondrial assays do not fully replicate in vivo physiological conditions, as they lack systemic factors such as blood flow, cellular interactions, and intact tissue architecture. However, these assays are well-established and widely accepted in the field for evaluating mitochondrial function under controlled conditions (Caspersen et al., 2008; Niatsetskaya et al., 2012). Despite their limitations, they enable direct comparisons of mitochondrial activity across experimental groups and provide valuable mechanistic insights that complement in vivo observations.

(5) PAM technique limits the resolution of the image beyond 500-750 micron depth. Assessing basal ganglia may not be possible with this approach?

We thank the reviewer for this important comment. We agree that the imaging depth of PAM is limited and may not allow assessment of deeper brain structures such as the basal ganglia. However, in our neonatal HI model—as in many clinical cases of HIE—cortical injury is typically more severe and represents a major focus for mechanistic and therapeutic investigations. The cortical regions assessed with PAM are thus highly relevant to the pathophysiology of neonatal HI. We have now acknowledged this depth limitation in the third paragraph of the newly added Limitations in this study subsection of the Discussion on page 15:

“Another limitation of this study is the restricted imaging depth of the PAM technique, which is typically less than 1 mm and therefore does not allow assessment of deeper brain structures such as the basal ganglia. However, in both our neonatal HI model and most clinical cases of neonatal hypoxia-ischemia, cortical injury tends to be more prominent and functionally significant. As such, our cortical measurements remain highly relevant for investigating the mechanisms of injury and evaluating therapeutic interventions.”

(6) Hypothermia in present study reduces the brain temperature from 37 to 29-32 degree centigrade. In clinical set up, head temp is reduced to 33-34.5 in neonatal hypoxia ischemia. Hence a drop in temperature to 29 degrees is much lower relative to the clinical practice. How the present study with greater drop in head temperature can be interpreted for understanding the pathophysiology of therapeutic hypothermia in neonatal HIE. Moreover, in HIE model using higher temperature of 37 and dropping to 29 seems to be much different than the clinical scenario. Please discuss.

We thank the reviewer for raising this important point regarding temperature ranges in our study. In Figure 1, we used a broader temperature range (down to 29°C) to explore the general relationship between temperature and CMRO_2_ in uninjured neonatal mice. This experiment was not intended to model therapeutic hypothermia directly, but rather to characterize the baseline physiological responses.

For all experiments involving hypothermia as a therapeutic intervention following HI, we consistently maintained a brain temperature of 32°C, which falls within the clinically accepted mild hypothermia range for neonatal HIE (typically 33–34.5°C). We believe this temperature closely mimics clinical practice and supports the translational relevance of our findings.

(7) NMR was assessed ex-vivo. How does it relate to in vivo assessment. Infants admitted in Neonatal intensive Care Unit, frequently get MRI with spectroscopy. How do the MRS findings in human newborns with HIE correlate with the ex-vivo evaluation of metabolites.

We thank the reviewer for this insightful question. While our study assessed brain metabolites ex vivo, similar metabolic changes have been observed in vivo using proton magnetic resonance spectroscopy (¹H-MRS) in infants with HIE. Specifically, reductions in N-acetylaspartate (NAA) — a marker of neuronal integrity — have been reported in neonates with severe brain injury, aligning with our ex vivo findings. This correlation between in vivo and ex vivo assessments supports the translational relevance of our model for studying metabolic disruption in neonatal HIE. We have added this point to the subsection Using Optically Measured CMRO_2_ to Detect Neonatal HI Brain Injury of the Results on page 8, along with a supporting clinical reference (Lally et al., 2019):

“In addition, in vivo proton MRS in infants with HIE has also shown a reduction in NAA, particularly in cases of severe injury (Lally et al., 2019). This reduction in NAA, observed in neonatal intensive care settings, reflects neuronal and axonal loss or dysfunction and serves as a biomarker for injury severity. The alignment between our ex vivo observations and in vivo MRS findings in clinical studies reinforces the translational relevance of our model for investigating metabolic disturbances in neonatal HIE.”

**Reviewer #3 (Public review)**
(1) In Sun et al. present a comprehensive study using a novel photoacoustic microscopy setup and mitochondrial analysis to investigate the impact of hypoxia-ischemia (HI) on brain metabolism and the protective role of therapeutic hypothermia. The authors elegantly demonstrate three connected findings: (1) HI initially suppresses brain metabolism, (2) subsequently triggers a metabolic surge linked to oxidative phosphorylation uncoupling and brain damage, and (3) therapeutic hypothermia mitigates HI-induced damage by blocking this surge and reducing mitochondrial stress.The study's design and execution are great, with a clear presentation of results and methods. Data is nicely presented, and methodological details are thorough.

We thank the reviewer for the positive feedback.

(2) However, a minor concern is the extensive use of abbreviations, which can hinder readability. As all the abbreviations are introduced in the text, their overuse may render the text hard to read to non-specialist audiences. Additionally, sharing the custom Matlab and other software scripts online, particularly those used for blood vessel segmentation, would be a valuable resource for the scientific community. In addition, while the study focuses on the short-term effects of HI, exploring the long-term consequences and definitively elucidating HI's impact on mitochondria would further strengthen the manuscript's impact.

We thank the reviewer for these valuable suggestions. Please find our point-by-point responses below:

Abbreviations: To improve readability, we have added a List of Abbreviations on page 3 to help readers, especially non-specialists, navigate the terminology more easily.

MATLAB Code Availability: The methodology for blood vessel segmentation was described in detail in our previous publication (Sun et al., 2020). We have now updated the subsection Quantification of Cerebral Hemodynamics and Oxygen Metabolism by PAM of the Methods on page 18 to provide additional details and have indicated that the MATLAB scripts are available upon request.

“Briefly, this process involves generating a vascular map using signal amplitude from the Hilbert transformation, selecting a region slightly larger than the vessel of interest, and applying Otsu’s thresholding method to remove background pixels. Isolated or spurious boundary fragments are then removed to improve boundary smoothness. The customized MATLAB code used for vessel segmentation is available upon request.”

Long-Term Effects of Hypothermia: We agree that exploring long-term outcomes would enhance the broader impact of this research. While our study focuses on the acute phase following HI, prior studies have shown long-term neuroprotective benefits of therapeutic hypothermia, such as enhanced white matter development (Koo et al., 2017). We have added this point to the fourth paragraph in the subsection Limitations in this study of the Discussion on page 15:

“While our study focuses on the acute effects of hypothermia, previous research has shown long-term neuroprotective benefits, including improved white matter development post-injury (Koo et al., 2017). These findings highlight hypothermia's potential for both immediate and extended recovery, warranting further study of long-term outcomes.”

(3) Extensive use of abbreviations.

Thank you for the helpful suggestion. To improve readability for a broader audience, we have added a List of Abbreviations on page 3 of the manuscript to assist readers in navigating terminology used throughout the text. This has been included as Response #2 to Reviewer #3.

(4) Share code used to conduct the study.

Thank you for the suggestion. The methodology for vessel segmentation was previously published (Sun et al., 2020), and we have noted in the subsection Quantification of Cerebral Hemodynamics and Oxygen Metabolism by PAM of the Methods on page 18 that the MATLAB code is available upon request. This has also been included as Response #2 to Reviewer #3.

Reference:

Cao R, Li J, Kharel Y, Zhang C, Morris E, Santos WL, Lynch KR, Zuo Z, Hu S. 2018. Photoacoustic microscopy reveals the hemodynamic basis of sphingosine 1-phosphate-induced neuroprotection against ischemic stroke. Theranostics 8:6111–6120. doi:10.7150/thno.29435

Caspersen CS, Sosunov A, Utkina-Sosunova I, Ratner VI, Starkov AA, Ten VS. 2008. An Isolation Method for Assessment of Brain Mitochondria Function in Neonatal Mice with Hypoxic-Ischemic Brain Injury. Developmental Neuroscience 30:319–324. doi:10.1159/000121416

Clancy B, Kersh B, Hyde J, Darlington RB, Anand KJS, Finlay BL. 2007. Web-based method for translating neurodevelopment from laboratory species to humans. Neuroinformatics 5:79–94. doi:10.1385/ni:5:1:79

Greenberg RS, Zahurak M, Belden C, Tunkel DE. 1998. Assessment of oropharyngeal distance in children using magnetic resonance imaging. Anesth Analg 87:1048–1051. doi:10.1097/00000539-199811000-00014

Kiyatkin EA. 2007. Brain temperature fluctuations during physiological and pathological conditions. Eur J Appl Physiol 101:3–17. doi:10.1007/s00421-007-0450-7

Koo E, Sheldon RA, Lee BS, Vexler ZS, Ferriero DM. 2017. Effects of therapeutic hypothermia on white matter injury from murine neonatal hypoxia-ischemia. Pediatr Res 82:518–526. doi:10.1038/pr.2017.75

Lally PJ, Montaldo P, Oliveira V, Soe A, Swamy R, Bassett P, Mendoza J, Atreja G, Kariholu U, Pattnayak S, Sashikumar P, Harizaj H, Mitchell M, Ganesh V, Harigopal S, Dixon J, English P, Clarke P, Muthukumar P, Satodia P, Wayte S, Abernethy LJ, Yajamanyam K, Bainbridge A, Price D, Huertas A, Sharp DJ, Kalra V, Chawla S, Shankaran S, Thayyil S, MARBLE consortium. 2019. Magnetic resonance spectroscopy assessment of brain injury after moderate hypothermia in neonatal encephalopathy: a prospective multicentre cohort study. Lancet Neurol 18:35–45. doi:10.1016/S1474-4422(18)30325-9

Lin W, Powers WJ. 2018. Oxygen metabolism in acute ischemic stroke. J Cereb Blood Flow Metab 38:1481–1499. doi:10.1177/0271678X17722095

Mallard C, Vexler Z. 2015. Modeling ischemia in the immature brain: how translational are animal models? Stroke 46:3006–3011. doi:10.1161/STROKEAHA.115.007776

Niatsetskaya ZV, Sosunov SA, Matsiukevich D, Utkina-Sosunova IV, Ratner VI, Starkov AA, Ten VS. 2012. The Oxygen Free Radicals Originating from Mitochondrial Complex I Contribute to Oxidative Brain Injury Following Hypoxia–Ischemia in Neonatal Mice. J Neurosci 32:3235–3244. doi:10.1523/JNEUROSCI.6303-11.2012

Sheldon RA, Windsor C, Ferriero DM. 2018. Strain-Related Differences in Mouse Neonatal Hypoxia-Ischemia. Dev Neurosci 40:490–496. doi:10.1159/000495880

Sun N, Bruce AC, Ning B, Cao R, Wang Y, Zhong F, Peirce SM, Hu S. 2022. Photoacoustic microscopy of vascular adaptation and tissue oxygen metabolism during cutaneous wound healing. Biomed Opt Express, BOE 13:2695–2706. doi:10.1364/BOE.456198

Sun N, Ning B, Bruce AC, Cao R, Seaman SA, Wang T, Fritsche-Danielson R, Carlsson LG, Peirce SM, Hu S. 2020. In vivo imaging of hemodynamic redistribution and arteriogenesis across microvascular network. Microcirculation 27:e12598. doi:10.1111/micc.12598

Sun N, Zheng S, Rosin DL, Poudel N, Yao J, Perry HM, Cao R, Okusa MD, Hu S. 2021. Development of a photoacoustic microscopy technique to assess peritubular capillary function and oxygen metabolism in the mouse kidney. Kidney International 100:613–620. doi:10.1016/j.kint.2021.06.018